# 🦙 AHA: A VISION-LANGUAGE-MODEL FOR DETECTING AND REASONING OVER FAILURES IN ROBOTIC MANIPULATION

**Jiafei Duan**[1,2]    **Wilbert Pumacay**[3]    **Nishanth Kumar**[1,4]
**Yi Ru Wang**[1,2]    **Shulin Tian**[5]    **Wentao Yuan**[1,2]
**Ranjay Krishna**[2,6]    **Dieter Fox**[1,2]    **Ajay Mandlekar**[*1]    **Yijie Guo**[*1]
[1]NVIDIA,    [2]University of Washington,    [3]Universidad Católica San Pablo,    [4]MIT
[5]Nanyang Technological University,    [6]Allen Institute for Artificial Intelligence

## ABSTRACT

Robotic manipulation in open-world settings requires not only task execution but also the ability to detect and learn from failures. While recent advances in vision-language models (VLMs) and large language models (LLMs) have improved robots' spatial reasoning and problem-solving abilities, they still struggle with failure recognition, limiting their real-world applicability. We introduce AHA, an open-source VLM designed to detect and reason about failures in robotic manipulation using natural language. By framing failure detection as a free-form reasoning task, AHA identifies failures and provides detailed, adaptable explanations across different robots, tasks, and environments. We fine-tuned AHA using `FailGen`, a scalable framework that generates the first large-scale dataset of robotic failure trajectories, the AHA dataset. `FailGen` achieves this by procedurally perturbing successful demonstrations from simulation. Despite being trained solely on the AHA dataset, AHA generalizes effectively to real-world failure datasets, robotic systems, and unseen tasks. It surpasses the second-best model (GPT-4o in-context learning) by 10.3% and exceeds the average performance of six compared models including five state-of-the-art VLMs by 35.3% across multiple metrics and datasets. We integrate AHA into three manipulation frameworks that utilize LLMs/VLMs for reinforcement learning, task and motion planning, and zero-shot trajectory generation. AHA 's failure feedback enhances these policies' performances by refining dense reward functions, optimizing task planning, and improving sub-task verification, boosting task success rates by an average of 21.4% across all three tasks compared to GPT-4 models. Project page: aha-vlm.github.io.

## 1    INTRODUCTION

In recent years, foundation models have made remarkable progress across various domains, demonstrating their ability to handle open-world tasks (Driess et al., 2023; Alayrac et al., 2022; Achiam et al., 2023; Zhang et al., 2023). These models, including large language models (LLMs) and vision-language models (VLMs), have shown proficiency in interpreting and executing human language instructions (Ouyang et al., 2022), producing accurate predictions and achieving strong task performance. However, despite these advancements, key challenges remain—particularly with hallucinations, where models generate responses that deviate from truth. Unlike humans, who can intuitively detect and adjust for such errors, these models often lack the mechanisms for recognizing their own mistakes (Lin et al., 2021; Chen et al., 2021; Heyman, 2008).

Learning from failure is a fundamental aspect of human intelligence. Whether it's a child learning to skate or perfecting a swing, the ability reason over failures is essential for improvement (Young, 2009; Gopnik, 2020; Heyman, 2008). The concept of improvement through failures is widely applied in training foundation models and is exemplified by techniques such as Reinforcement Learning with

---

[*]Equal advising

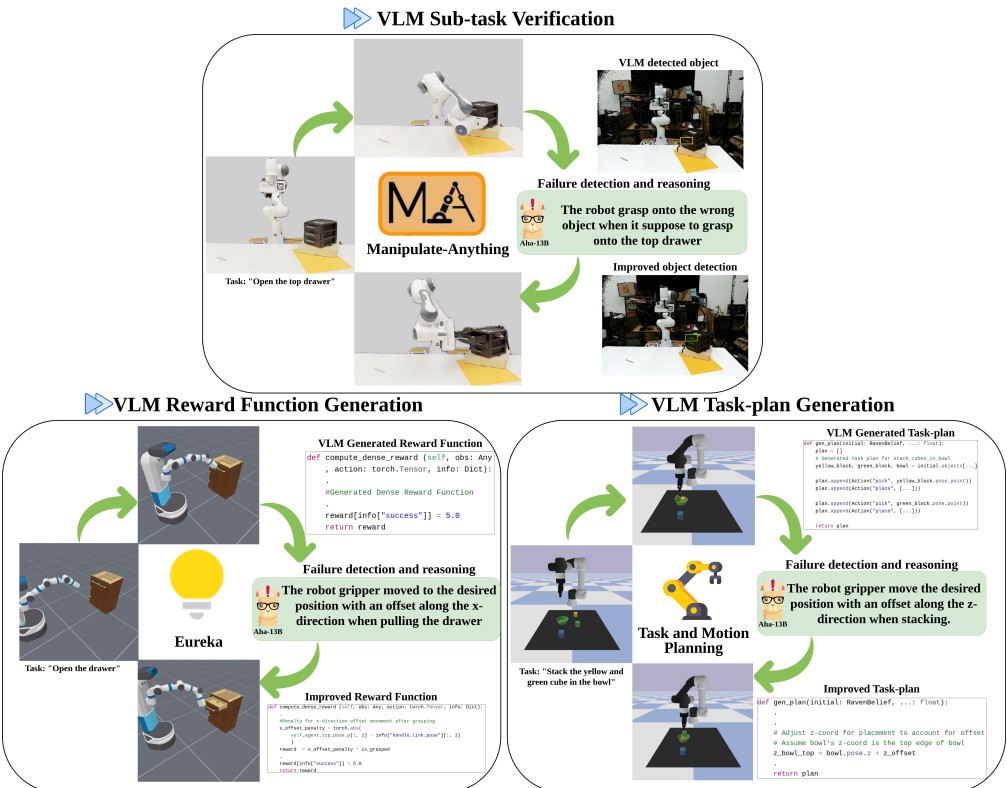

Figure 1: AHA is a Vision-Language Model designed to detect and reason about failures in robotic manipulation. As an instruction-tuned VLM, it can enhance task performance in robotic applications that utilize VLMs for reward generation, task planning, or sub-task verification. By incorporating AHA into the reasoning pipeline, these applications can achieve accelerated and improved performance.

Human Feedback (RLHF) (Ouyang et al., 2022; Christiano et al., 2017), where human oversight and feedback steers models toward desired outcomes. This feedback loop plays a critical role in aligning generative models with real-world objectives. However, a crucial question persists: How can we enable these models to autonomously detect and reason about their own failures, particularly in robotics, where interactions and environments are stochastic and unpredictable?

This need is particularly pressing in robotics, where foundation models such as VLMs and LLMs are increasingly used to address open-world tasks. Recent advancements have enabled these models to tackle spatial reasoning, object recognition, and multimodal problem-solving—skills vital for robotic manipulation (Reid et al., 2024; OpenAI, 2024; Yuan et al., 2024; Chen et al., 2024; Wang et al., 2023b). VLMs and LLMs are already being integrated to automate reward generation for reinforcement learning (Ma et al., 2023; 2024), develop task plans for motion planning (Curtis et al., 2024), and even generate zero-shot robot trajectories (Huang et al., 2023; 2024a; Duan et al., 2024; Huang et al., 2024b). While these models excel at task execution, they often face challenges in detecting and reasoning over failures—skills that are crucial for navigating dynamic and complex environments. For example, if a robot drops an object mid-task, a human observer would immediately recognize the error and take corrective action. How can we empower robots with similar capabilities, allowing them not only to perform tasks but also to detect and learn from their mistakes?

To learn from their mistakes, robots must first detect and understand why they failed. We introduce AHA, an open-source VLM that uses natural language to detect and reason about failures in robotic manipulation. Unlike prior work that treats failure reasoning as a binary detection problem, we frame it as a free-form reasoning task, offering deeper insights into failure mode reasoning. Our model not only identifies failures but also generates detailed explanations. This approach enables AHA to adapt to various robots, camera viewpoints, tasks, and environments in both simulated and real-world

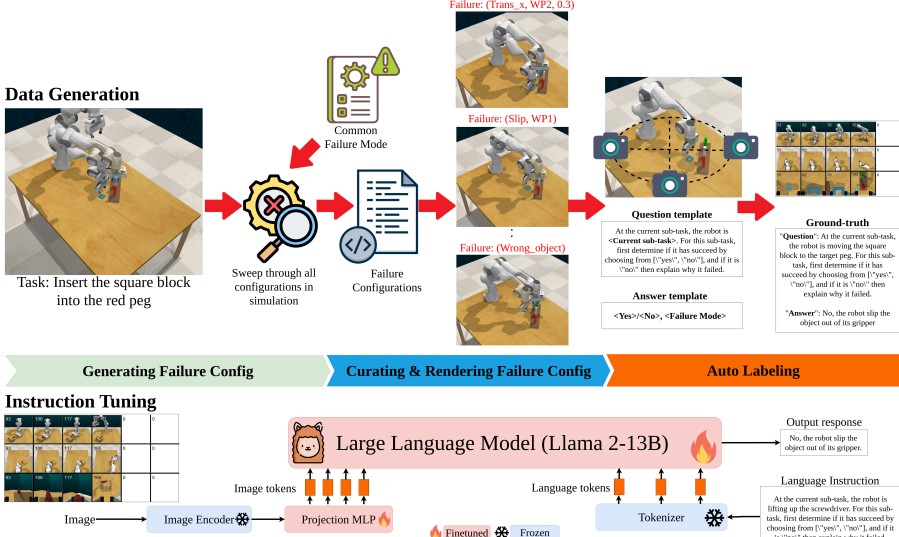

Figure 2: **Overview of AHA Pipeline**. (Top) The data generation for AHA is accomplished by taking a normal task trajectory in simulation and procedurally perturbing all keyframes using our taxonomy of failure modes. Through `FailGen`, we systematically alter keyframes to synthesize failure demonstrations conditioned on the original tasks. Simultaneously, we generate corresponding query and answer prompts for each task and failure mode, which are used for instruction-tuning. (Bottom) The instruction-tuning pipeline follows the same fine-tuning procedure as LLaVA-v1.5 Liu et al. (2023a), where we fine-tune only the LLM base model—in this case, LLaMA-2-13B and the projection linear layers, while freezing the image encoder and tokenizer.

scenarios. It can also be integrated into downstream robotic applications leveraging VLMs and LLMs, shown in Figure 1. We make the following three major contributions:

**1. We introduce `FailGen`, a data generation pipeline for the procedural generation of failure demonstration data for robotic manipulation tasks across simulators.** To instruction-tune AHA, we developed `FailGen`, the first automated data generation pipeline that procedurally creates the AHA dataset—a large-scale collection of robotic manipulation failures with over 49K+ image-query pairs across 79 diverse simulated tasks. Despite being fine-tuned only on the AHA dataset, AHA demonstrates strong generalization to real-world failure datasets, different robotic systems, and unseen tasks, as evaluated on three separate datasets not included in the fine-tuning. `FailGen` is also flexible data generation pipeline integrates seamlessly with various simulators, enabling scalable procedural generation of failure demonstrations.

**2. We demonstrate that AHA excels in failure reasoning, generalizing across different embodiments, unseen environments, and novel tasks, outperforming both open-source and proprietary VLMs.** Upon fine-tuning AHA, we benchmarked it against six state-of-the-art VLMs, both open-source and proprietary, evaluating performance across four metrics on three diverse evaluation datasets, each featuring different embodiments, tasks, and environments out-of-distribution from the training data. AHA outperformed GPT-4o model by more than 20.0% on average across datasets and metrics, and by over 43.0% compared to LLaVA-v1.5-13B (Liu et al., 2023a), the base model from which AHA is derived. This demonstrates AHA's exceptional ability to detect and reason about failures in robotic manipulation across embodiment and domains.

**3. We show that AHA enhances downstream robotic applications by providing failure reasoning feedback.** We demonstrate that AHA can be seamlessly integrated into robotic applications that utilize VLMs and LLMs. By providing failure feedback, AHA improves reward functions through Eureka reflection, enhances task and motion planning, and verifies sub-task success in zero-shot robotic manipulation. Across three downstream tasks, our approach achieved an average success rate 21.4% higher than GPT-4 models, highlighting AHA's effectiveness in delivering accurate natural language failure feedback to improve task performance through error correction.

## 2 RELATED WORK

AHA enables language reasoning for failure detection in robotic manipulation, enhancing downstream robotics applications. To provide context, we review progress in: 1) failure detection in robotic manipulation, 2) data generation in robotics, and 3) foundation models for robotic manipulation.

**Failure Detection in Robotic Manipulation**. Failure detection and reasoning have long been studied in the Human-Robot Interaction (HRI) community (Ye et al., 2019; Khanna et al., 2023) and in works leveraging Task and Motion Planning (TAMP) (Garrett et al., 2020). With the recent widespread adoption of LLMs and VLMs in robot manipulation systems—either for generating reward functions or synthesizing robot trajectories (Ma et al., 2023; 2024) in a zero-shot manner—the importance of detecting task failures has regained prominence (Huang et al., 2023; Duan et al., 2024; Skreta et al., 2024; Ha et al., 2023; ?). Most modern approaches focus on using off-the-shelf VLMs or LLMs as success detectors (Ma et al., 2022; Ha et al., 2023; Wang et al., 2023a; Duan et al., 2024; Dai et al., 2024), and some employ instruction-tuning of VLMs to detect failures (Du et al., 2023).Furthermore, hallucinations often occur in LLMs and VLMs. Methods that leverage these models for failure detection can mitigate this issue by detecting uncertainty in VLMs, as demonstrated in this work Zheng et al. (2024). However, these methods are often limited to binary success detection and does not provide language explanations for why failures occur. Our framework introduces failure reasoning in a new formulation, generating language-based explanations of failures to aid robotics systems that leverage VLMs and LLMs in downstream tasks. Additionally, we investigated whether AHA suffers from hallucinations by analyzing the prediction probabilities of sentence tokens. We found that AHA exhibits fewer hallucinations compared to other VLMs (see supplementary material).

**Data Generation in Robotics** There have been many methods in robotic manipulation that automate data generation of task demonstrations at scale (Mandlekar et al., 2023; Hoque et al., 2024), whether for training behavior cloning policies, instruction-tuning VLMs (Yuan et al., 2024), or curating benchmarks for evaluating robotic policies in simulation (Xie et al., 2024; Pumacay et al., 2024). A well-known example is MimicGen (Mandlekar et al., 2023), which automates task demonstration generation via trajectory adaptation by leveraging known object poses. Additionally, works like RoboPoint use simulation to generate general-purpose representations for robotic applications, specifically for fine-tuning VLMs. Similarly, systems like The Colosseum Pumacay et al. (2024) automate data generation for curating benchmarks in robotic manipulation. Our approach aligns closely with RoboPoint, as we also leverage simulation to generate data for instruction-tuning VLMs. However, unlike RoboPoint, we focus on synthesizing robotic actions in simulation rather than generating representations like bounding boxes or points.

**Foundation Models for Robotic Manipulation.** In recent years, leveraging foundation models for robotic manipulation has gained significant attention due to the effectiveness of LLMs/VLMs in interpreting open-world semantics and their ability to generalize across tasks (Duan et al., 2022; Hu et al., 2023; Firoozi et al., 2023; Urain et al., 2024). Two main approaches have emerged: the first uses VLMs and LLMs in a promptable manner, where visual prompts guide low-level action generation based on visual inputs (Liu et al., 2024a; Huang et al., 2024a;b). The second focuses on instruction-tuning VLMs for domain-specific tasks (Li et al., 2024). For example, RoboPoint (Yuan et al., 2024) is tuned for spatial affordance prediction, and Octopi (Yu et al., 2024) for physical reasoning using tactile images. These models generalize beyond their training data and integrate seamlessly into manipulation pipelines. Our approach follows this second path, developing a scalable method for generating instruction-tuning data in simulation and fine-tuning VLMs specialized in detecting and reasoning about robotic manipulation failures, with applications that extend beyond manipulation tasks to other robotic domains.

## 3 THE AHA DATASET

We leveraged `FailGen` to procedurally generate the AHA dataset from RLBench tasks (James et al., 2020) and used it for the instruction-tuning of AHA. In this section, we begin by categorizing common failure modes in robotics manipulation and defining a taxonomy of failures in Section 3.1. Next, we explain how this taxonomy is used with `FailGen` to automate the data generation for the AHA dataset in simulation in Section 3.2.

Table 1: **AHA datasets for instruction-tuning.** We combined the AHA dataset, our large-scale robotic manipulation failure dataset, with VQA and object detection data. By incorporating this diverse data mix into the fine-tuning process, AHA is able to reason about failures in robotic manipulation across different domains, embodiments, and tasks.

| Source | The AHA dataset (Train) | VQA (Liu et al., 2023a) | LVIS (Gupta et al., 2019) |
|---|---|---|---|
| |  |  |  |
| Quantity | 49K | 665K | 100K |
| Query | For the given sub-tasks, first determine it has succeed by choosing from ["yes", "no"] and then explain the reason why the current sub-tasks has failed. | What is the cat doing in the image? | Find all instances of drawer. |
| Answer | No, The robot gripper rotated with an incorrect roll angle | The cat is sticking its head into a vase or container, possibly drinking water or investigating the interior of the item. | [(0.41, 0.68, 0.03, 0.05), (0.42, 0.73, 0.04, 0.08), ...] |

## 3.1 FAILURE MODES IN ROBOTIC MANIPULATION

To curate an instruction-tuning dataset of failure trajectories for robotic manipulation tasks, we began by systematically identifying prevalent failure modes. Our approach involved a review of existing datasets, including DROID (Khazatsky et al., 2024) and Open-X Embodiment (Padalkar et al., 2023), as well as an analysis of policy rollouts from behavior cloning models. We examined failures occurring in both teleoperated and autonomous policies. Building upon prior works, such as REFLECT (Liu et al., 2023d), we formalized a taxonomy encompassing seven distinct failure modes commonly observed in robotic manipulation: incomplete grasp, inadequate grip retention, misaligned keyframe, incorrect rotation, missing rotation, wrong action sequence, and wrong target object.

**Incomplete Grasp (`No_Grasp`) Failure:** `No_Grasp` is an object-centric failure that occurs when the gripper reaches the desired grasp pose but fails to close before proceeding to the next keyframe.

**Inadequate Grip Retention (`Slip`) Failure:** `Slip` is an object-centric failure that happens after the object has been successfully grasped. As the gripper moves the object to the next task-specific keyframe, the grip loosens, causing the object to slip from the gripper.

**Misaligned keyframe (`Translation`) Failure:** This action-centric failure occurs when the gripper moves toward a task keyframe, but a translation offset along the X, Y, or Z axis causes the task to fail with respect to a fixed reference coordinate system.

**Incorrect Rotation (`Rotation`) Failure:** `Rotation` occurs when the gripper reaches the correct position but rotates to an incorrect angle in roll, pitch, or yaw relative to a fixed reference point. Although it attempts the rotation, the misalignment due to rotation results in task failure.

**Missing Rotation (`No_Rotation`) Failure:** `No_Rotation` occurs when the gripper reaches the correct position but fails to perform the necessary rotation in roll, pitch, or yaw. The absence of any rotation when it is required leads to misalignment and ultimately causes the task to fail.

**Wrong Action Sequence (`Wrong_action`) Failure:** `Wrong_action` is an action-centric failure that occurs when the robot executes actions out of order, performing an action keyframe before the correct one. For example, in the task `put_cube_in_drawer`, the robot moves the cube toward the drawer before opening it, leading to task failure.

**Wrong Target Object (`Wrong_object`) Failure:** `Wrong_object` is an object-centric failure that occurs when the robot acts on the wrong target object, not matching the language instruction. For example, in the task `pick_the_red_cup`, the gripper picks up the green cup, causing failure.

## 3.2 Implementation of the Aha dataset

The AHA dataset is generated with RLBench James et al. (2020), utilizing its keyframe-based formulation to dynamically induce failure modes during task execution. RLBench natively provides keyframes for task demonstrations, which enables flexibility in object manipulation (handling tasks with varying objects) and the sequence of actions (altering the execution order of keyframes). Building on this foundation, we leverage `FailGen`, our custom environment wrapper around RLBench that allows for task-specific trajectory modifications through keyframes perturbations, object substitutions, and reordering of keyframe sequences. `FailGen` systematically generates failure trajectories aligned with the taxonomy defined in Section 3.1, yielding a curated dataset of 49k failure-question pairs.

To generate the AHA dataset, we systematically sweep through all keyframes in each RLBench task, considering all potential configurations of the seven failure modes that could result in overall task failure. By leveraging the success condition checker in the simulation, we procedurally generate `YAML`-based configuration files by sweeping through each failure mode across all keyframes. These files provide details on potential failure modes, parameters (such as distance, task sequence, gripper retention strength, etc.), and corresponding keyframes that `FailGen` should perturb to induce failure. Additionally, we incorporate language templates to describe what the robot is doing between consecutive keyframes. Using these descriptions along with the failure modes, we can systematically curate question-answer pairs for each corresponding failure mode.

For specific failure modes, `No_Grasp` is implemented by omitting gripper open/close commands at the relevant keyframes, effectively disabling gripper control. `Slip` introduces a timed release of the gripper shortly after activation. `Translation` and `Rotation` perturb the position and orientation of a keyframe, respectively, while `No_Rotation` constrains the keyframe's rotational axis. `Wrong_Action` reorders keyframe activations to simulate incorrect sequencing, and `Wrong_Object` reassigns the keyframes intended for one object to another, maintaining the relative pose to mimic improper object manipulation. Using this pipeline, we also successfully generated a failure dataset from ManiSkill (Tao et al., 2024) and adapted RoboFail (Liu et al., 2023d) for the evaluation of AHA. This further demonstrates the generalizability and versatility of `FailGen` in generating failure cases across different simulation environments.

## 4 Method

This section outlines the failure reasoning problem formulation (Sec.4.1) used to fine-tune and evaluate AHA. Next, we discuss the curated data mix used for co-finetuning AHA (Sec.4.2). Finally, we detail the instruction fine-tuning pipeline and the model architecture selection for AHA (Sec.4.3).

### 4.1 Failure Reasoning Formulation

We extend prior work (Skreta et al., 2024; Duan et al., 2024) by introducing a two-step framework for robot failure analysis that combines sub-task success detection and failure reasoning. Sub-task success is evaluated as a binary classification problem (*Yes/No*), while failure reasoning is performed using vision-language models (VLMs) to generate natural language explanations for the causes of failure. This approach allows for both precise failure detection and interpretability in robot manipulation tasks. Manipulation tasks are represented as trajectories consisting of a sequence of sub-tasks $\{S_0, S_1, \ldots, S_T\}$, where each sub-task $S_t$ is defined by two consecutive keyframes $(K_t, K_{t+1})$. Each sub-task corresponds to an atomic manipulation action, such as "grasping a cube" in a stacking task. For each sub-task, the input to the VLM includes a query prompt and a structured image representation. The query prompt is generated using a template specific to the sub-task and describes the task context and success condition.

The image input is represented as a matrix $\mathbf{I} \in \mathbb{R}^{n \times T \times H \times W \times C}$, where rows correspond to camera viewpoints $\{V_0, V_1, \ldots, V_{n-1}\}$ and columns correspond to temporal keyframes $\{K_0, K_1, \ldots, K_T\}$. To capture the spatiotemporal progression of the task, frames are arranged in temporal order, and missing keyframes are replaced with white patches. We include several camera viewpoints to mitigate occlusions and ensure a comprehensive spatial context. This combined representation enables the VLM to reason over the robot's trajectory and diagnose failure causes effectively, as demonstrated in Table 1.

## 4.2 SYNTHETIC DATA FOR INSTRUCTION-TUNING

To facilitate the instruction-tuning of AHA, we needed to systematically generate failure demonstration data. To achieve this, we developed `FailGen`, an environment wrapper that can be easily applied to any robot manipulation simulator. `FailGen` systematically perturbs successful robot trajectories for manipulation tasks, transforming them into failure trajectories with various modes of failure as depicted in Figure 2 (Top image). Using `FailGen`, we curated the AHA dataset (Train) dataset by alternating across 79 different tasks in the RLBench simulator, resulting in 49k failure image-text pairs. Furthermore, following proper instruction-tuning protocols for VLMs (Liu et al., 2023a) and building on prior works (Brohan et al., 2023; Yuan et al., 2024), co-finetuning is crucial to the success of instruction fine-tuning of VLMs. Therefore, in addition to the AHA dataset, we co-finetuned AHA with general visual question-answering (VQA) datasets sourced from internet data, which helps models retain pre-trained knowledge. Specifically, we included the VQA dataset (Liu et al., 2023a), containing 665k conversation pairs, and the LVIS dataset (Gupta et al., 2019), which comprises 100k instances with predicted bounding box centers and dimensions, as summarized in Table 1.

## 4.3 INSTRUCTION FINE-TUNING

We followed the instruction-tuning pipeline outlined by (Liu et al., 2023b). As depicted in Fig. 2, our model architecture includes an image encoder, a linear projector, a language tokenizer, and a transformer-based language model. The image encoder processes images into tokens, projected by a 2-layer linear projector into the same space as the language tokens. These multimodal tokens are then concatenated and passed through the language transformer. All components are initialized with pre-trained weights. During fine-tuning, only the projector and transformer weights are updated, while the vision encoder and tokenizer remain frozen. The model operates autoregressively, predicting response tokens and a special token marking the boundary between instruction and response.

## 4.4 IMPACT ON DOWNSTREAM TASKS

AHA integrates failure reasoning to address limitations in downstream robotics methods, improving reward synthesis, decision-making, and feedback efficiency. In reinforcement learning (RL), AHA refines reward synthesis by analyzing rollouts to provide failure explanations, enabling iterative adjustments to dense reward functions and improving sample efficiency, as demonstrated in approaches such as Eureka (Ma et al., 2023). In task and motion planning (TAMP) systems like PRoC3S (Curtis et al., 2024), AHA enhances feedback loops by interpreting visualizations of failed plans, generating failure explanations, and informing language-model-based plan refinement. This process improves robustness in long-horizon tasks by addressing semantic errors overlooked by finite failure checks. In open-ended frameworks like Manipulate-Anything (Duan et al., 2024), AHA improves subtask verification by analyzing sequential frames for task progression errors, reducing failure propagation in zero-shot data generation. These integrations enable systematic reasoning improvements across RL, TAMP, and data generation, directly enhancing task success and robustness.

## 5 EXPERIMENTAL RESULTS

In this section, we evaluate AHA's detection and reasoning performance against six state-of-the-art VLMs, including both open-source and proprietary models, some utilizing in-context learning. The evaluation spans three diverse datasets, covering out-of-domain tasks, various simulation environments, and cross-embodiment scenarios. We then assess AHA's ability to retain general world knowledge after fine-tuning on domain-specific data. Finally, we explore its potential to improve downstream robotic manipulation applications.

## 5.1 EXPERIMENTAL SETUP

To quantitatively evaluate AHA's detection and reasoning capabilities for failures in robotic manipulation, we curated two failure datasets and adapted an existing failure dataset for benchmarking. To ensure a fair comparison of free-form language reasoning, we also employed four different evaluation metrics to measure semantic similarity between sentences.

Table 2: **Quantitative Evaluation on Failure Detection and Reasoning.** AHA-13B was evaluated and benchmarked against three open and three proprietary VLMs and one visual prompting baseline across three evaluation datasets. AHA-13B outperformed all other VLMs on every evaluation dataset and nearly every evaluation metric, with the exception of the AHA (Test) dataset, where GPT-4o exceeded by less than 3%.

| Models | AHA dataset (Test set) | | | | ManiSkill-Fail | | | | REFLECT | | | |
|---|---|---|---|---|---|---|---|---|---|---|---|---|
| | ROUGE$_L$ ↑ | Cos Sim ↑ | BinSucc(%) ↑ | Fuzzy Match ↑ | ROUGE$_L$ ↑ | Cos Sim ↑ | BinSucc(%) ↑ | Fuzzy Match ↑ | ROUGE$_L$ ↑ | Cos Sim ↑ | BinSucc(%) ↑ | Fuzzy Match ↑ |
| LLaVA-v1.5-13B (Liu et al., 2023a) | 0.061 | 0.208 | 0.080 | 0.648 | 0.000 | 0.208 | 0.022 | 0.270 | 0.000 | 0.203 | 0.000 | 0.404 |
| LLaVA-NeXT-34B (Liu et al., 2024b) | 0.013 | 0.231 | 0.017 | 0.626 | 0.001 | 0.195 | 0.007 | 0.277 | 0.018 | 0.188 | 0.017 | 0.351 |
| Qwen-VL (Bai et al., 2023) | 0.000 | 0.161 | 0.000 | 0.426 | 0.037 | 0.301 | 0.116 | 0.034 | 0.000 | 0.159 | 0.000 | 0.050 |
| Gemini-1.5 Flash (Reid et al., 2024) | 0.120 | 0.231 | 0.371 | 0.566 | 0.003 | 0.121 | 0.014 | 0.032 | 0.000 | 0.042 | 0.000 | 0.393 |
| GPT-4o | 0.251 | 0.308 | 0.500 | **0.784** | 0.142 | 0.335 | 0.688 | 0.453 | 0.114 | 0.318 | 0.554 | 0.438 |
| GPT-4o-ICL (5-shot) | 0.226 | 0.380 | 0.611 | 0.776 | 0.341 | 0.429 | 0.971 | 0.630 | 0.236 | 0.429 | 0.571 | 0.418 |
| AHA-7B | 0.434 | 0.574 | 0.691 | 0.695 | **0.609** | 0.680 | 1.000 | 0.532 | 0.204 | 0.394 | 0.625 | 0.439 |
| AHA-13B (Ours) | **0.446** | **0.583** | **0.702** | 0.768 | 0.600 | **0.681** | 1.000 | **0.633** | **0.280** | **0.471** | **0.643** | **0.465** |

Table 3: **Quantitative Evaluation on Standard VQA Benchmarks.** AHA-13B performs on par with LLaVA-13B Liu et al. (2023a), the VLM from which AHA adapts its fine-tuning strategy.

| | MMBench | ScienceQA | TextVQA | POPE | VizWiz |
|---|---|---|---|---|---|
| | (Liu et al., 2023c) | (Lu et al., 2022) | (Singh et al., 2019) | (Li et al., 2023) | (Gurari et al., 2018) |
| LLaVA-13B (LLama-2) (Liu et al., 2023a) | **67.70** | **73.21** | **67.40** | **88.00** | 53.01 |
| AHA-13B (LLama-2) | 65.20 | 71.94 | 65.20 | 85.74 | **53.45** |

**Benchmarks.** We curated three datasets to evaluate AHA's reasoning and failure detection capabilities, benchmarking against other state-of-the-art VLMs. The first dataset, AHA dataset (Test), includes 11k image-question pairs from 10 RLBench tasks, generated similarly to the fine-tuning data via `FailGen` (Section 3.2) but without overlapping with the finetuning dataset. It evaluates AHA's ability to generalize to novel, out-of-domain tasks. The second dataset, ManiSkill-Fail, comprises 130 image-question pairs across four tasks in ManiSkill (Tao et al., 2024), generated using `Failgen` wrapper on the ManiSkill simulator. This dataset assesses AHA's performance in a different simulator and under changing viewpoints. Lastly, we adapted a failure benchmark from the RoboFail dataset (Liu et al., 2023d), which features real-world robot failures in seven UR5 robot tasks, allowing for evaluation across simulation, real-world trajectories, and different embodiments.

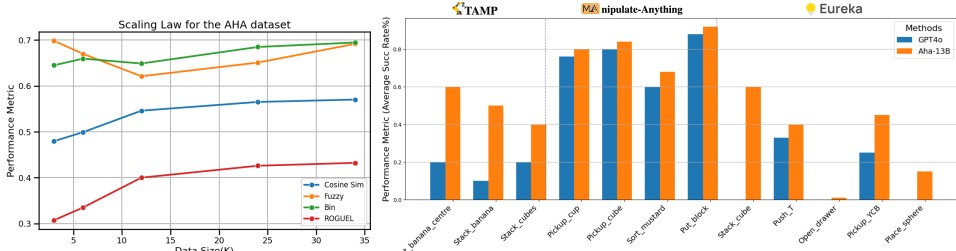

Figure 3: (Left) **Scaling law with the AHA dataset**. Scaling of effect of model performance with varying domain specific fine-tuning data. (Right) **Downstream Robotic Application Performance.** AHA-13B outperforms GPT-4o in reasoning about failures within these robotic applications, leading to improved performance of the downstream tasks.

**Evaluation Metrics.** To fairly evaluate success detection and language reasoning across all datasets and baselines, we employ four metrics. First, the **ROUGE-L score** measures the quality of generated text by focusing on the longest common subsequence between candidate and reference texts. Second, we use **Cosine Similarity** to assess similarity between texts or embeddings, avoiding the "curse of dimensionality". Third, **LLM Fuzzy Matching** utilizes an external language model—specifically, Anthropic's unseen model, `claude-3-sonnet`—to evaluate semantic similarity in a teacher-student prompting format (Zhou et al., 2023). Lastly, we calculate a **Binary success rate** by comparing the model's predictions directly against the ground truth for success detection.

## 5.2 QUANTITATIVE EXPERIMENTAL RESULTS

We contextualize the performance of AHA by conducting a systematic evaluation of failure reasoning and detection across these three datasets, general VQA datasets, and performed ablation studies.

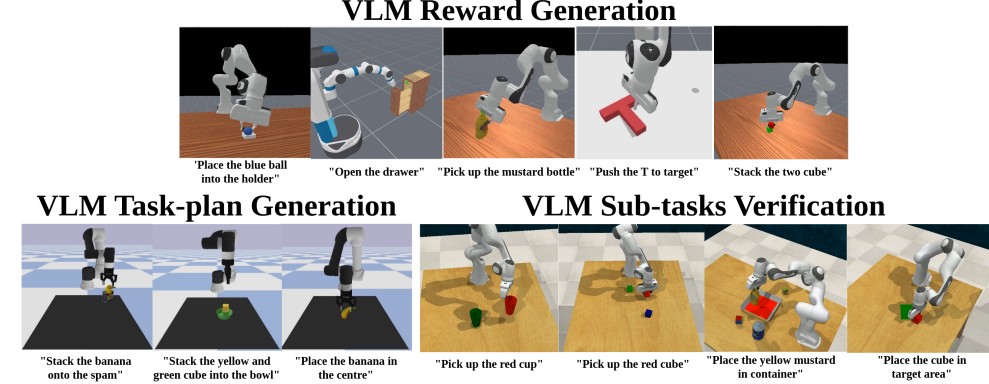

Figure 4: **Downstream Robotic Application.** We demonstrated that AHA can be integrated into existing LLM/VLM-assisted robotic applications to provide failure reasoning and feedback, helping to accelerate and improve task success rates in these systems.

**AHA generalizes across embodiments, unseen environments, and novel tasks.** To ensure fairness and eliminate bias in the detection and reasoning capabilities of AHA, we evaluated it on three different datasets that were never seen during fine-tuning, each designed to test a specific form of generalization. First, on the AHA dataset (test) dataset, AHA demonstrated its ability to **generalize reasoning across tasks and new behaviors within the same domain, outperforming the second-best performing VLM, GPT-4o ICL**, by an average margin of 12.6% difference across all evaluation metrics. Second, we assessed AHA-13B on a dataset generated by the `Failgen` wrapper in a **different simulation domain**, ManiSkill, showing that our model outperforms GPT-4o-ICL by an average of 13.4% difference across all metrics as depicted in Table 2. Lastly, to demonstrate **generalization to real-world robots and different embodiments**, we evaluated AHA-13B on RoboFail (Liu et al., 2023d), where it outperforms GPT-4o-ICL by 4.9% difference.

**AHA retains common sense knowledge.** We evaluated AHA-13B's performance on various VQA benchmarks and present the results in Table 3 . AHA-13B **performs comparably to LLaVA-v1.5-13B (LLama-2) (Liu et al., 2023a)** , with only a 1.5% margin difference as depicted in Table 3. Notably, LLaVA-v1.5-13B is a VLM trained on the same pre-trained weights as AHA-13B but fine-tuned on VQA data. This indicates that AHA-13B is capable of functioning as a general purpose VLM, in addition to excelling at failure reasoning.

**AHA's performance scales with data size.** We evaluated Aha's performance using a range of AHA data for instruction fine-tuning, spanning [3k, 6k, 12k, 34k, 48k, 60k], and co-trained individual checkpoints corresponding to these data sizes as shown in Figure 3 (Left). The model was then assessed on the ManiSkill-Fail dataset across four evaluation metrics. An average quadratic fit of 0.0022 across all four metrics demonstrates a **scaling effect with fine-tuning on our procedurally generated data pipeline**. This suggests that further scaling can improved model performance.

## 5.3 Downstream Robotics Tasks

We demonstrate that AHA's failure detection and reasoning capabilities are useful across a wide spectrum of downstream robotics applications. This includes automatic reward generation for reinforcement learning applications (Ma et al., 2023), automatic task plan generation for task and motion planning applications (Curtis et al., 2024), and as an improved verification step for automatic data generation systems (Duan et al., 2024).

**AHA enables efficient reward synthesis for reinforcement learning.** To evaluate this downstream task, we adapted Eureka's (Ma et al., 2023) implementation to the ManiSkill simulator, which offers more state-based manipulation tasks. We strictly followed the Eureka reward function generation and reflection pipeline, modifying it by incorporating perception failure feedback via either AHA-13B or GPT-4o (acting as a baseline) to enhance the original LLM reflection mechanism. Instead of only including a textual summary of reward quality based on policy training statistics for automated reward editing, we further incorporated explanations of policy failures based on evaluation rollouts. We

evaluated our approach on five reinforcement learning tasks from ManiSkill, ranging from tabletop to mobile manipulation. To systematically assess the reasoning capabilities of different VLMs under budget constraints, we sampled one reward function initially and allowed for iterations over two sessions of GPT API calls. Each policy was trained using PPO over task-specific training steps and evaluated across 1,000 test steps. During policy rollouts, we employed either AHA-13B or GPT-4o for reward reflection to improve the reward function. Comparing the evaluated policy success rates using different failure feedback VLMs, we observed that AHA-13B provided intuitive, human-level failure reasoning that aided in modifying and improving generated dense reward functions. This resulted in success across all five tasks within the budget constraints, and our approach **outperformed GPT-4o by a significant margin of 22.34% in task success rate** shown in Figure 3 (Right).

**AHA refines task-plan generation for TAMP.** To demonstrate AHA's utility within a planning system, we incorporated our approach into PRoC3S (Curtis et al., 2024). The PRoC3S system solves tasks specified in natural language by prompting an LLM for a Language-Model Program (LMP) that generates plans, and then testing a large number of these plans within a simulator before executing valid plans on a robot. If no valid plan can be found within a certain number of samples (100 in our experiments), the LLM is re-prompted for a new LMP given failure information provided by the environment. Importantly, as is typical of TAMP methods, the original approach checks for a finite set of failures (inverse kinematics, collisions, etc.) from the environment, and returns any sampled plan that does not fail in any of these ways. We incorporated a VLM into this pipeline in two ways: (1) we prompt the VLM with visualizations of failed plan executions within the simulator, ask it to return an explanation for the failure, and feed this back to PRoC3S' LLM during the LMP feedback stage, (2) after PRoC3S returns a valid plan, we provide a visualization of this to the VLM and ask it to return whether this plan truly achieves the natural language goal, with replanning triggered if not. We compared GPT-4o and AHA-13B as the VLM-based failure reasoning modules within this implementation of PRoC3S across three tasks (shown in Figure 4). Each task was evaluated over 10 trials, with a maximum of 100 sampling steps and three feedback cycles provided by either GPT-4o or AHA-13B. The success rate for each task was recorded. As shown in Figure Figure 3 (Right), utilizing AHA-13B for **failure reasoning significantly improved the task success rate and outperforming GPT-4o by a substantial margin of 36.7%**.

**AHA improves task verification for zero-shot robot data generation.** To demonstrate AHA's utility in zero-shot robot demonstration generation, we integrated our approach into the `Manipulate-Anything` framework. This open-ended system employs various Vision-Language Models (VLMs) to generate diverse robot trajectories and perform a wide range of manipulation tasks without being constrained by predefined actions or scenarios. A critical component of `Manipulate-Anything` is its sub-task verification module, which analyzes past and current frames to decide whether a sub-task has been achieved before proceeding or re-iterating over the previous sub-task. We replaced the original VLM (GPT-4V) in the sub-task verification module with AHA-13B and evaluated performance across four RLBench tasks (Figure 4), conducting 25 episodes for each task. Our results show that **substituting the sub-task verification module's VLM with AHA improved reasoning accuracy and overall task success by an average of 5%**.

## 6 CONCLUSION

**Limitations.** AHA currently outputs language reasoning that is closely aligned with the failure scenarios in the fine-tuning data. However, there is an opportunity to output more open-ended failures, to cover those arising from modes outside of the ones included in the failure taxonomy. Additionally, while `FailGen` systematically curates failure data from simulations, distilling large pretrained policies to perform diverse tasks in simulation and sampling failure modes would allow us to generate more open-ended failure examples, potentially enhancing the instruction-tuned performance of AHA.

**Conclusion.** We introduce AHA, an open-source vision-language model that significantly enhances robots' ability to detect and reason about manipulation task failures using natural language. By framing failure detection as a free-form reasoning task, AHA not only identifies failures but also provides detailed explanations adaptable to various robots, tasks, and environments. Leveraging `FailGen` and the curated AHA dataset, we trained AHA on a diverse set of robotic failure trajectories. Our evaluations show that AHA outperforms existing models across multiple metrics and datasets. When integrated into manipulation frameworks, its natural language feedback greatly improves error recovery and policy performance compared to GPT-4 models. These results demonstrate AHA's effectiveness in enhancing task performance through accurate error detection and correction.

ACKNOWLEDGMENTS

This project is done as part of Jiafei Duan's NVIDIA internship project. Jiafei Duan is supported by the Agency for Science, Technology and Research (A*STAR) National Science Fellowship. Wilbert Pumacay is supported by grant 234-2015-FONDECYT from Cienciactiva of the National Council for Science, Technology and Technological Innovation (CONCYTEC-PERU). I would also like to extend my gratitude to all members of the NVIDIA Seattle Robotics Lab for their invaluable suggestions and feedback.

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

# 7 APPENDIX

## 7.1 OVERVIEW

The Appendix contains the following content.

- **Failure Taxonomy** (Appendix 7.2): more thorough definition and figure to discussions about the different failure modes.
- **FailGen Data Generation Pipeline** (Appendix 7.3): more discussion of FailGen implementation with example configurations files.
- **AHA Datasets** (Appendix 7.4): more details on the instruction-tuning dataset and evaluation datasets.
- **Additional Experimental Results** (Appendix 7.5): more details and experiments with instruction finetuning.
- **Downstream Robotic Application: VLM Reward Generation** (Appendix 7.6): more policy rollouts, generated reward function examples, and prompts.
- **Downstream Robotic Application: VLM Task-plan Generation**(Appendix 7.7): more policy rollouts, generated task-plan examples, and prompts.
- **Downstream Robotic Application: VLM Sub-task Verification**(Appendix 7.8): more policy rollouts.

## 7.2 FAILURE TAXONOMY

We conducted an in-depth study of recent real-world, diverse robot datasets (such as Open-X (Padalkar et al., 2023), DROID (Khazatsky et al., 2024), and EGO4D (Grauman et al., 2022)) and the policies trained using these datasets. Through this analysis, we identified several common modes of failure, which can be categorized into seven types: incomplete grasp, inadequate grip retention, misaligned keyframe, incorrect rotation, missing rotation, wrong action sequence, and wrong target object.

**Incomplete Grasp (`No_Grasp`) Failure:** `No_Grasp` is an object-centric failure that occurs when the gripper reaches the desired grasp pose but fails to close before proceeding to the next keyframe.

**Inadequate Grip Retention (`Slip`) Failure:** `Slip` is an object-centric failure that occurs after the object has been successfully grasped. As the gripper moves the object toward the next task-specific keyframe, the grip weakens, causing the object to slip from the gripper. For generating the AHA dataset for training and evaluation, we configured a 5-timestep activation for the `Slip` failure mode, triggering the object to drop from the gripper.

**Misaligned keyframe (`Translation`) Failure:** This action-centric failure occurs when the gripper moves toward a task keyframe, but a translation offset along the X, Y, or Z axis causes the task to fail. For the AHA training and evaluation dataset, we introduced a translation offset of [-0.5, 0.5] meters. In the ManiSkill-Fail dataset, we applied a translation noise of [0, 0.1] meters along either the X, Y, or Z axis from the original waypoint. The translation coordinate system is depicted in Figure 7 (Left).

**Incorrect Rotation (`Rotation`) Failure:** `Rotation` is an action-centric failure that occurs when the gripper reaches the desired translation pose for the sub-task keyframe, but there is an offset in roll, yaw, or pitch, leading to task failure. For the AHA dataset, we set a rotation offset of [-3.14, 3.14] in radians along roll, yaw, or pitch. The rotation coordinate system is depicted in Figure 7 (Right).

**Missing Rotation (`No_Rotation`) Failure:** `No_Rotation` is an action-centric failure that happens when the gripper reaches the desired translation pose but fails to achieve the necessary rotation (roll, yaw, or pitch) for the sub-task, resulting in task failure.

**Wrong Action Sequence (`Wrong_action`) Failure:** `Wrong_action` is an action-centric failure that occurs when the robot executes actions out of order, performing an action keyframe before the correct one. For example, in the task `put_cube_in_drawer`, the robot moves the cube toward the drawer before opening it, leading to task failure.

**Wrong Target Object (`Wrong_object`) Failure:** `Wrong_object` is an object-centric failure that occurs when the robot acts on the wrong target object, not matching the language instruction. For

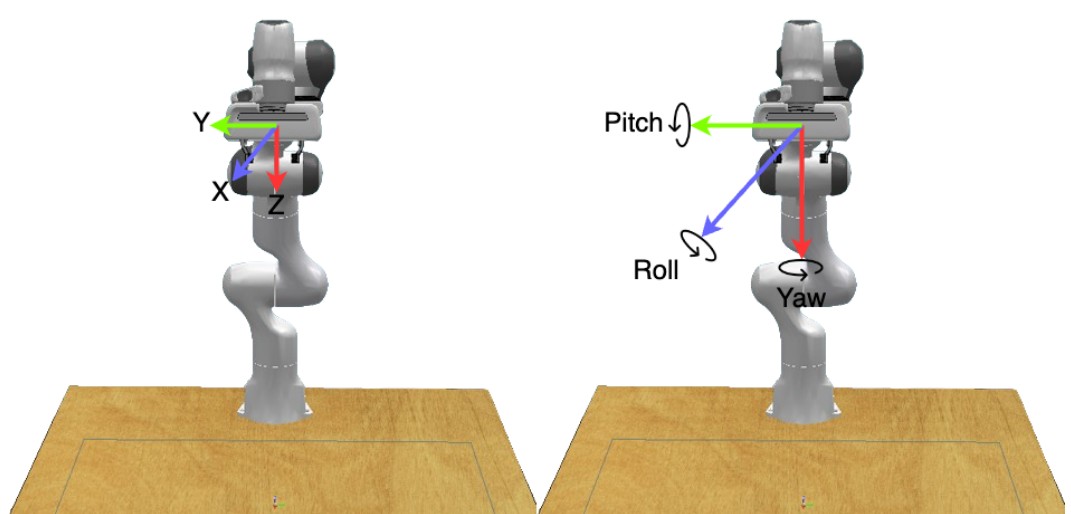

Figure 5: **Failure mode reference coordinate systems.** (Left) Translation coordinate system, and (Right) rotation coordinate system.

example, in the task `pick_the_red_cup`, the gripper picks up the green cup, causing failure. We perform a sweep through all manipulable objects, swapping them with the target object in the scene.

### 7.3 FAILGEN DATA GENERATION PIPELINE

We developed `FailGen`, an environment wrapper that can be easily integrated into any simulator. It leverages pre-defined or hand-crafted robot demonstrations for imitation learning, where each trajectory is represented as a waypoint-based system. Two consecutive waypoints form a sub-task, with each sub-task linked to a predefined set of language descriptions. `FailGen` allows for modifications to environment parameters, such as gripper end-effector translation, rotation, and open/close state. By altering these parameters, we systematically generate failures at every waypoint. However, for the 79 tasks collected from RLBench, we do not initially know which sub-task will fail due to specific failure modes. To address this, we perform a systematic sweep, using RLBench's built-in success conditions to explore all possible combinations. This generates a configuration of failures for each task, which we then use to procedurally generate all failure training data. Additionally, we manually annotate each sub-task with natural language instructions describing the task, and pair this with failure mode explanations to serve as language input for instruction-tuning. Example of the configuration files are depicted at Figure 9.

### 7.4 AHA DATASET

Using `FailGen`, we curated two datasets from RLBench (James et al., 2020). The first is the training dataset, AHA dataset (train), which is used for instruction-tuning AHA alongside the co-train dataset. The second is the testing dataset, AHA dataset (test), used for evaluation. AHA dataset (train) contains approximately 49k image-query pairs of failures derived from 79 tasks, while AHA dataset (test) consists of around 11k image-query pairs from 10 hold-out tasks.

### 7.5 ADDITIONAL EXPERIMENTAL RESULTS

We conducted additional experiments to better understand and visualize AHA's predictions. We trained two versions of the AHA model with 13B parameters, using different language models for fine-tuning: Llama-2-13B and Vicuna-1.5-13B. The results showed less than a 2.5% performance difference between the two models, indicating that our fine-tuning data is effective regardless of the base language model. These results are presented in Table 6. Additionally, we visualized the output

```
1   save_path: /home/${oc.env:USER}/data/failgen_data
2   obs_mode: rgb
3   render_mode: sensors
4   shader: default
5   sim_backend: auto
6   image_size: [256, 256]
7   stages: [0, 1, 2, 3]
8   failures:
9     - type: grasp
10      enabled: false
11      stages: [2]
12    - type: trans_x
13      enabled: false
14      stages: [0, 1, 3]
15      noise: 0.1
16    - type: trans_y
17      enabled: false
18      stages: [0, 1, 3]
19      noise: 0.1
20    - type: trans_z
21      enabled: false
22      stages: [0, 1, 3]
23      noise: 0.1
24
```

```
data:
  # Where to save the demos
  save_path: /home/data
  # The size of the images to save
  waypoints: [0, 1, 2, 3]
failures:
  - type: grasp
    name: failure_grasp_pose
    enabled: False
    waypoints: [1]

  - type: translation_y
    name: trans_y
    enabled: False
    waypoints: [1,2,3]
    range: [-0.5, 0.5]

  - type: rotation_x
    name: rot_x
    enabled: False
    waypoints: [0]
    range: [-1.57, 1.57]

  - type: wrong_sequence
    name: bad_seq
    enabled: False
    waypoints: [2,3]

sub-tasks:
  - task_no: 0
    enabled: False
    type: dummy
    targets: [ball]
    processes: [waypoint0, waypoint1]
    task_description: [
      "grasp onto the clock knob",
      "pick on the clock knob"
    ]
  - task_no: 1
    enabled: False
    type: dummy
    targets: [ball]
    processes: [waypoint1, waypoint2]
    task_description: [
      "rotate the knob",
      "turn the knob"
    ]
  - task_no: 2
    enabled: False
    type: dummy
    targets: [ball]
    processes: [waypoint2, waypoint3]
    task_description: [
      "let go",
      "release the gripper"
    ]
```

Figure 6: (Left) **Example of config file of one task for Maniskill-Fal**. (Right) **Example of config file for AHA task**

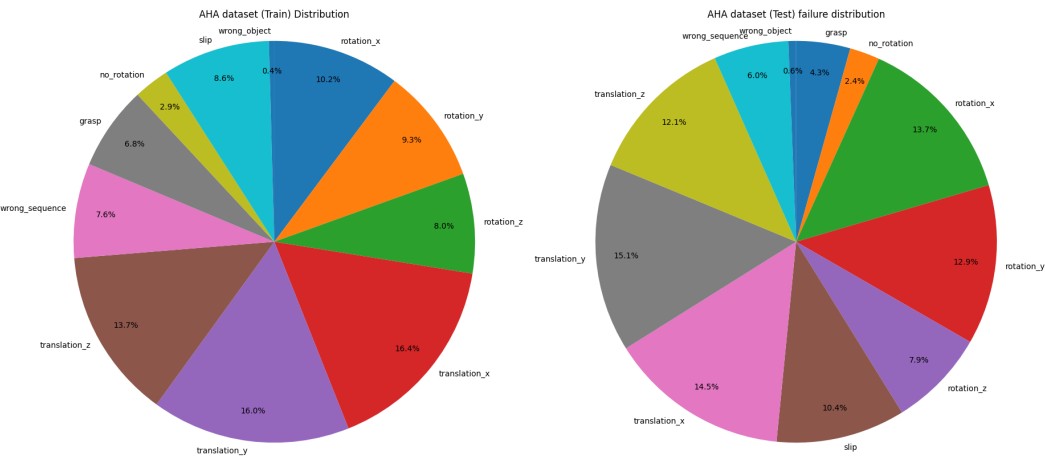

Figure 7: **Data distribution of AHA dataset for both train and test.**

predictions from various baselines compared to our model and evaluated performance across all three datasets, with the results shown in Figure 7.

**AHA model performance uncertainty estimation.** To evaluate the relationship between uncertainty estimation and model performance, we conducted additional experiments across various evaluation datasets. Specifically, we compared the sentence token prediction probabilities of AHA-13B with those of LLaVA v1.5-13B. AHA-13B exhibited significantly higher average prediction probabilities, reflecting its superior accuracy. These findings underscore the positive impact of fine-tuning with the AHA failure dataset on model confidence and performance as depicted in Table 5.

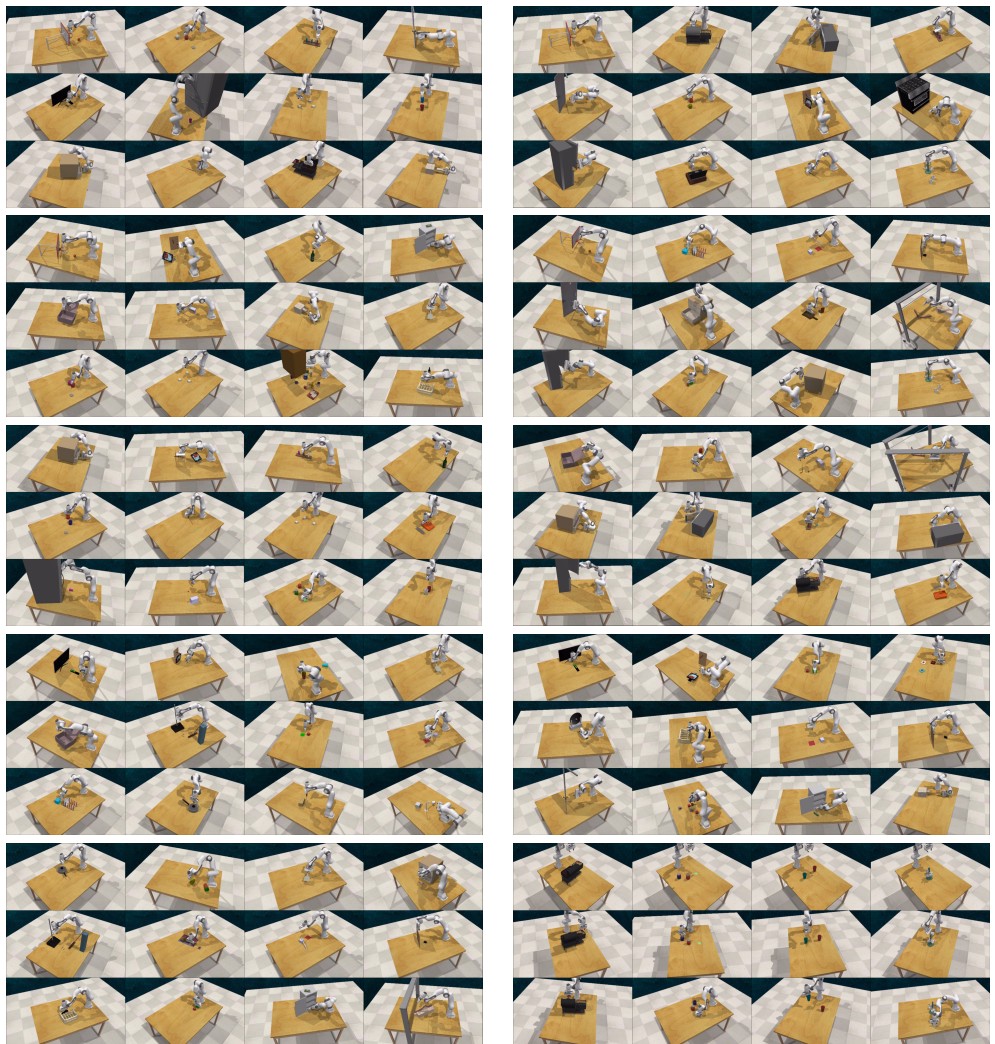

Figure 8: **Examples of different failure modes.** Row 1: `No_grasp` and `Rotation_x`. Row 2: `Rotation_y` and `Rotation_z`. Row 3: `Slip` and `Wrong_sequence`. Row 4: `Translation_x` and `Translation_y`. Row 5: `Translation_z` and `Wrong_object`.

**Effects of Viewpoints on Evaluation.** We evaluated the reasoning capabilities of the AHA model on the ManiSkill-Failure dataset across three different viewpoint configurations (one, two, and three viewpoints). Interestingly, we observed a slight performance advantage when using single-viewpoint images. We attribute this to the resolution limitations of the LLaVA-1.5 visual encoder (256x256), where single-viewpoint inputs offer clearer and more focused visual information for failure reasoning, as summarized in Table 4.

| Model: AHA-13B (Viewpoints) | Binary Success | ROUGE-L | LLM Fuzzy Match | Cosine Similarity |
|---|---|---|---|---|
| **One viewpoint** | 1.000 | 0.673 | 0.587 | 0.712 |
| **Two viewpoints** | 1.000 | 0.615 | 0.587 | 0.671 |
| **Three viewpoints** | 1.000 | 0.600 | 0.633 | 0.681 |

Table 4: Performance comparison across different numbers of viewpoints for AHA-13B

| Dataset | AHA-13B (Output Probabilities / Cosine Similarity) | LLaVA-13B-v1.5 (Output Probabilities / Cosine Similarity) |
|---|---|---|
| **AHA Dataset (Test)** | 0.670 / 0.583 | 0.066 / 0.208 |
| **Maniskill Fail** | 0.457 / 0.681 | 0.024 / 0.208 |
| **RoboFail** | 0.292 / 0.471 | 0.000 / 0.203 |

Table 5: Performance against model prediction sentence probabilities likelihood evaluated across datasets.

Table 6: **Ablation on Different Base LLMs for Fine-Tuning.** We fine-tuned AHA-13B using both LLaMA-2-13B and Vicuna-1.5-13B as base LLM models. The quantitative results show that the average performance difference between the two models is less than 2.5%, indicating that our failure formulation and the AHA dataset are effective regardless of the base model selection.

| Models | AHA dataset (Test) | | | | ManiSkill-Fail | | | | RoboFail | | | |
|---|---|---|---|---|---|---|---|---|---|---|---|---|
| | ROUGE$_L$ ↑ | Cos Sim ↑ | BinSucc(%) ↑ | Fuzzy Match ↑ | ROUGE$_L$ ↑ | Cos Sim ↑ | BinSucc(%) ↑ | Fuzzy Match ↑ | ROUGE$_L$ ↑ | Cos Sim ↑ | BinSucc(%) ↑ | Fuzzy Match ↑ |
| AHA-13B (Llama-2) | **0.446** | 0.583 | 0.702 | **0.768** | **0.600** | **0.681** | **1.000** | 0.633 | 0.280 | **0.471** | **0.643** | 0.465 |
| AHA-13B (Vicuna-1.5) | 0.458 | **0.591** | **0.709** | 0.695 | 0.574 | 0.657 | **1.000** | **0.851** | **0.290** | 0.468 | **0.661** | **0.605** |

## 7.6 VLM Reward Generation

In this section, we present reward functions generated by GPT-4o and AHA for comparison, as shown in Figure 9. Additionally, we demonstrate RL policy rollouts improved through AHA 's failure feedback across all five tasks along with all the final dense reward function modified by AHA shown in Figure 10 and 11. For all tasks, except **put_sphere_on_holder** (trained with PPO for 10M steps), PPO was trained for 25M steps prior to reflection and evaluation.

**Simulation task Details** We describe each of the 4 tasks in detail, along with their Maniskill variations and success condition.

### 7.6.1 PICKUP YCB

**Filename:** `pick_single_ycb.py`

**Task:** Pick up the single YCB object and lift it up to target height.

**Success Metric**: The object position is within goal_thresh (default 0.025) euclidean distance of the goal position.

### 7.6.2 PUSH T

**Filename:** `push_T.py`

**Task:** Push the T into the T shaped area.

**Success Metric**: The 3D T block covers at least 90

### 7.6.3 PLACE SPHERE

**Filename:** `place_sphere_v1.py`

**Task:** Pick up the sphere and place it into the bin.

**Success Metric**: the sphere is on top of the bin. That is, the sphere's xy-distance to the bin goes near 0, and its z-distance to the bin goes near the sphere radius + the bottom bin block's side length the object is static. That is, its linear and angular velocities are bounded with a small value the gripper is not grasping the object.

### 7.6.4 STACK CUBE

**Filename:** `stack_cube_v1.py`

**Task:** Pick up the red cube and stack it onto the green cube.

**Success Metric**: the red cube is on top of the green cube (to within half of the cube size), the red cube is static, the red cube is not being grasped by the robot (robot must let go of the cube).

Table 7: **Examples of detection and reasoning over failures by various VLMs.** We sampled one evaluation data point from each of the three evaluation datasets and compared the reasoning predictions generated by different state-of-the-art VLMs alongside our AHA-13B. Our model consistently provides the most semantically similar reasoning to the ground-truth answers for the given queries.

| Evaluation Datasets | AHA dataset (Test) | Maniskill-Fail | RoboFail (Liu et al., 2023d) |
|---|---|---|---|
| | | | |
| Question | At the current sub-task, the robot is picking up the hockey stick. First, determine whether it has succeeded by choosing from ["yes", "no"]. If not, explain why it failed? | At the current sub-task, the robot is picking up the red cube. First, determine whether it has succeeded by choosing from ["yes", "no"]. If not, explain why it failed? | At the current sub-task, the robot is placing the apple into the bowl. First, determine whether it has succeeded by choosing from ["yes", "no"]. If not, explain why it failed? |
| LLaVA 1.5-13B | The image displays a time-lapse sequence of a robotic arm grabbing a hockey stick. | Based on the image, it depicts a different sub-task of a robot grasping a red block. | For the pick-up task, i would say the robot arm has succeeded in picking. |
| GPT4o | No, the robot arm fails to hold the hockey stick securely. | Yes, the robot arm has successfully picked up the red block. | No, the robot gripper failed to pick up the bowl in the second frame. |
| Gemini 1.5 Flash | Yes. | Yes, Yes, Yes. | Frame 1, no the gripper is not grasping the bowl and move it. |
| AHA-13B (Ours) | No, the robot slip the object out of its gripper. | No, the robot gripper fails to close the gripper. | No, this is not the right action sequence for the task. |
| Ground-truth Answer | No, the robot slip the object out of its gripper. | No, the robot gripper fails to close. | No, this is not the right action sequence for the task. |

### 7.6.5 OPEN DRAWER

**Filename:** `open_cabinet_drawer_v1.py`

**Task:** Pull open the drawer.

**Success Metric**: The drawer is open at least 90% of the way, and the angular/linear velocities of the drawer link are small.

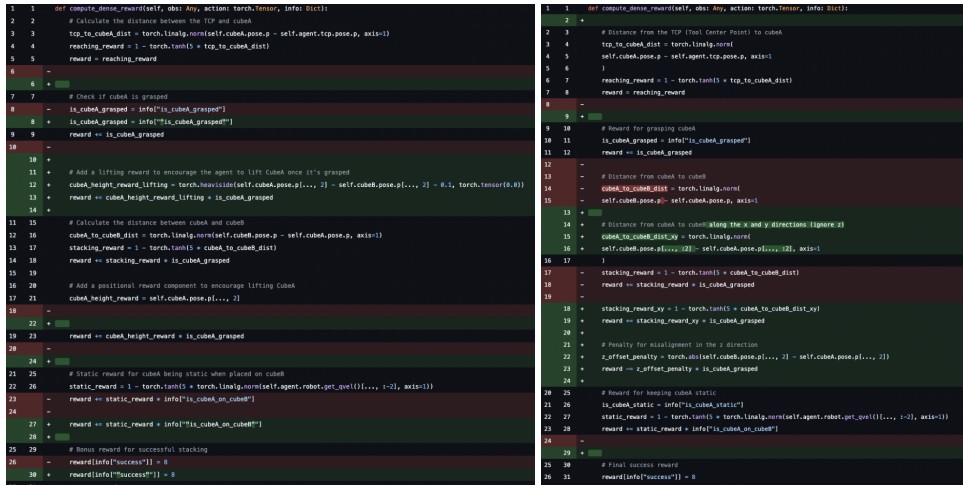

Figure 9: **(Left)** Example of improved dense reward function using GPT-4o for reflection. **(Right)** Example of improved dense reward function using AHA for reflection

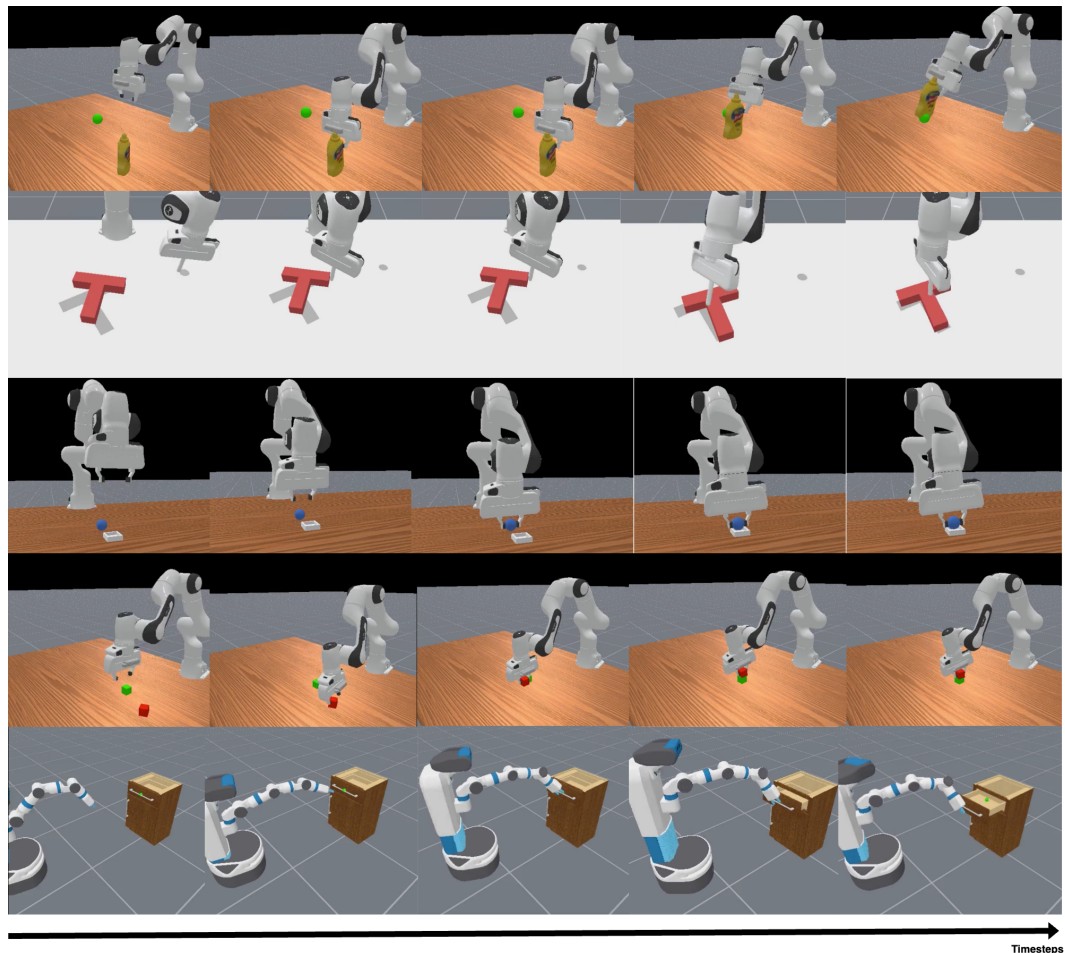

Figure 10: **RL policy roll-outs via improved with AHA.** Row 1: `pickup_YCB`. Row 2: `push_T`. Row 3: `Place_sphere`. Row 4: `stack_cube`. Row 5: `open_drawer`

## 7.7 VLM TASK-PLAN GENERATION

In this section, we present the policy rollouts improved by AHA in Figure 12, along with the modified task plans in Figure 13.

**Simulation task Details** We describe each of the 3 tasks in detail, along with their PyBullet variations and success condition.

### 7.7.1 PUT BANANA CENTRE

**Filename:** `ours_raven_ycb_pick.py`

**Task:** Pick up the banana and place it onto the centre of the table.

**Success Metric**: The success condition on the final location of the banana with respect to the table area.

### 7.7.2 STACK BANANA

**Filename:** `ours_ycb_banana_spam_stack.py`

**Task:** Pick up the banana and place it onto the spam can.

**Success Metric**: The position of the banana should be on the spam can, and rest stably.

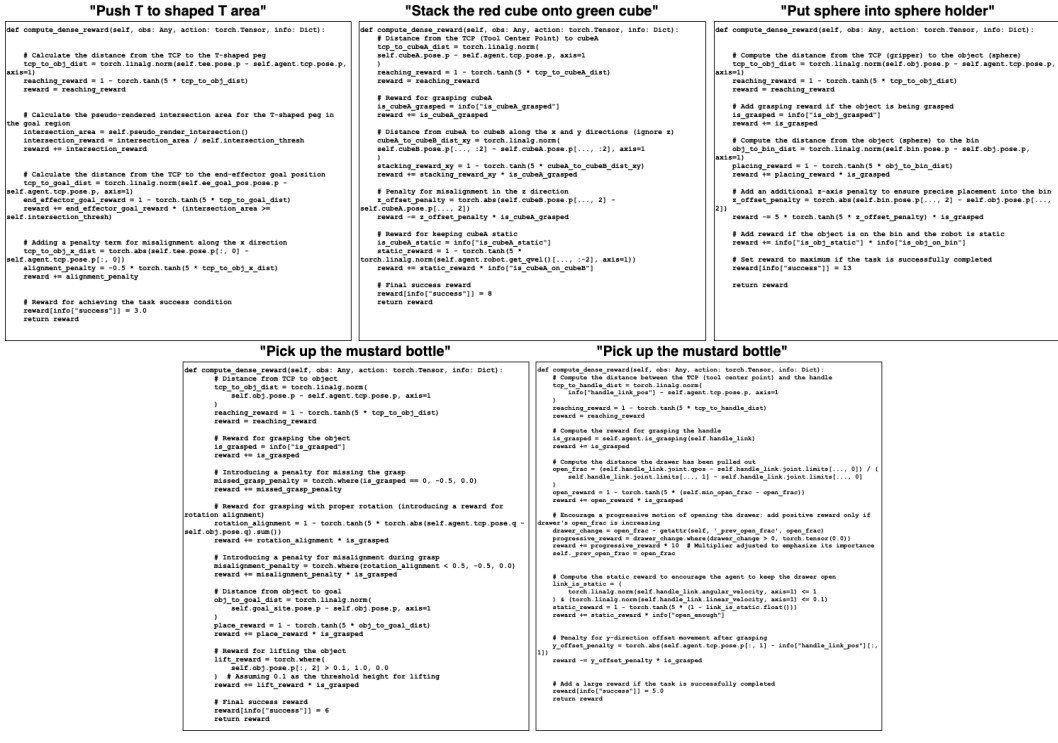

Figure 11: **Examples of modified reward function via AHA**

### 7.7.3 STACKS CUBES

**Filename:** `ours_raven_bowl_stack.py`

**Task:** Pick up the green cube and place into the green bowl, and then take the yellow cube and stack it on top of the green.

**Success Metric**: When the yellow cube is stably stack on top of the green in the green bowl.

### 7.8 VLM SUB-TASK VERIFICATION

In this section, we leverage `Manipulate-Anything` (Duan et al., 2024) as the main policy framework, integrating it with AHA. AHA functions as a sub-task verifier VLM, playing a crucial role in ensuring task success when using `Manipulate-Anything`. Examples of the roll-outs are shown in Figure 14.

**Simulation task Details** We describe each of the 4 tasks in detail, along with their RLBench variations and success condition.

### 7.8.1 PUT BLOCK

**Filename:** `put_block.py`

**Task:** Pick up the green block and place it on the red mat.

**Success Metric**: The success condition on the red mat detects the target green block.

### 7.8.2 PICKUP CUP

**Filename:** `pickup_cup.py`

**Task:** Pick up the red cup.

**Success Metric**: Lift up the red cup above the pre-defined location.

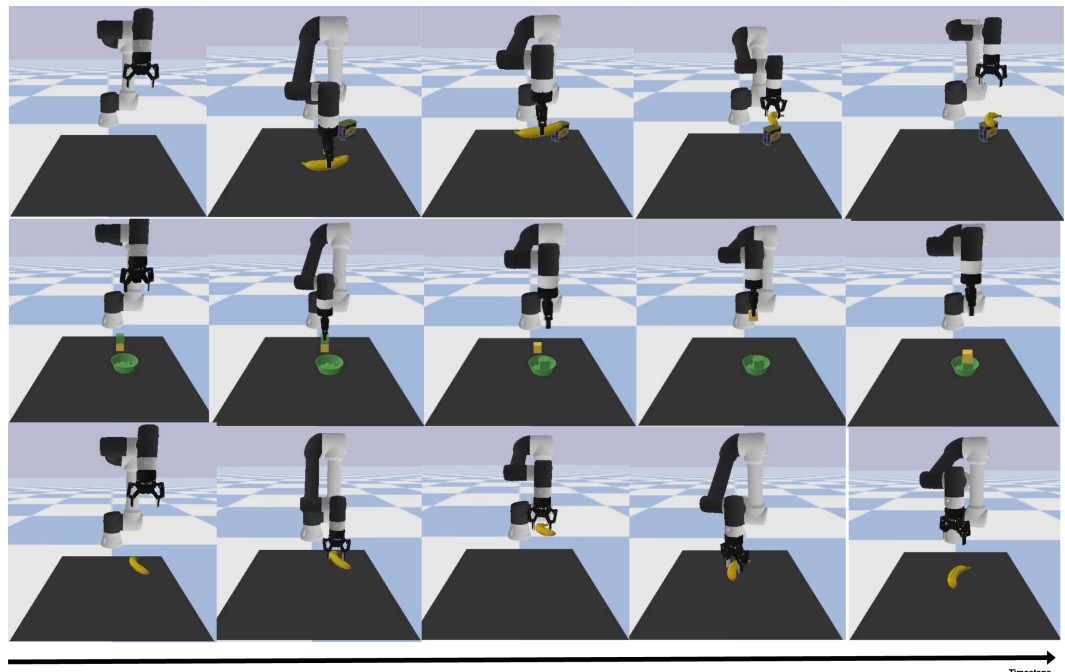

Figure 12: **TAMP policy roll-outs via improved with AHA.** Row 1: `put_banana_centre`. Row 2: `stack_banana`. Row 3: `stack_cubes`

### 7.8.3 SORT MUSTARD

**Filename:** `sort_mustard.py`

**Task:** Pick up the yellow mustard bottle, and place it into the red container.

**Success Metric**: The yellow mustard bottle inside red container.

### 7.8.4 PICK & LIFT

**Filename:** `pick_and_lift.py`

**Task:** Pick up the red cube.

**Success Metric**: The red cube is lifted up.

### 7.9 LIMITATIONS AND OPPORTUNITIES

While AHA is quite capable, it is not without limitations. In the following sections, we discuss some of these limitations and potential risks for real-world deployment.

**Does not cover all failure cases.** While AHA is instruction-tuned to address the seven most common failure modes in robotic tabletop manipulation, and provides fine-grained failure identification down to axis-level translation or orientation offsets of the gripper, it is inevitable that certain edge cases remain uncovered. These edge cases fall outside the distribution of failure modes considered in our approach and may occur in real-world scenarios. To address this limitation, future work could focus on distilling policies already trained with action-based "artifacts" to generate naturally occurring failures. These could then be labeled using human annotations or a hybrid approach combining AHA and a GPT-4o-based labeling system. Such efforts could enhance the robustness and applicability of failure detection across a broader range of scenarios.

**Improving policies directly with AHA** Currently, we have demonstrated AHA's effectiveness in enhancing existing robotics systems that utilize off-the-shelf VLMs/LLMs to automate various pipeline components. AHA has shown significant impact in improving code generation, task-plan

**"Stack banana onto spam can"**    **"Stack two cubes into a bowl"**

```
Thank you for the clarification. Given that the robot gripper moves to the desired position with an offset along the z direction,
we need to account for this offset when placing the banana on top of the potted meat can.
Let's assume the offset is a known constant, `GRIPPER_OFFSET_Z`. We'll use this offset to adjust the placement pose accordingly.
Here's the updated `gen_plan` and `gen_domain` functions:

```python
GRIPPER_OFFSET_Z = 0.02  # Example offset value, adjust as necessary

def gen_plan(initial: RavenBelief, grasp: RavenGrasp):
    plan = []

    # Pick up the banana
    plan.append(Action("pick", ["object_1", grasp]))

    # Get the pose of the potted meat can
    spam_pose = initial.objects["object_0"].pose

    # Calculate the placement pose for the banana on top of the potted meat can
    banana_place_pose = RavenPose(
        x=spam_pose.x,
        y=spam_pose.y,
        z=spam_pose.z + BLOCK_SIZE + GRIPPER_OFFSET_Z,  # Adjust for the gripper offset
        roll=spam_pose.roll,
        pitch=spam_pose.pitch,
        yaw=spam_pose.yaw
    )

    # Place the banana on top of the potted meat can
    plan.append(Action("place", ["object_1", grasp, banana_place_pose]))

    return plan

def gen_domain(initial:RavenBelief):
    return {
        "grasp": GraspSampler()
    }
```

### Explanation

1. **Picking the Banana**: The plan starts by picking up the banana (`object_1`) using a grasp sampled from the `GraspSampler`.
2. **Calculating Placement Pose**: The pose of the potted meat can (`object_0`) is retrieved from the initial state. The placement
pose for the banana is calculated by adding the height of the banana (approximated as `BLOCK_SIZE`) and the gripper offset
(`GRIPPER_OFFSET_Z`) to the z-coordinate of the potted meat can's pose. The roll, pitch, and yaw are kept the same to ensure proper
alignment.
3. **Placing the Banana**: The plan then places the banana at the calculated pose on top of the potted meat can.

### Why This Will Always Satisfy the Goal

- **Grasp Sampling**: The grasp is sampled from a `GraspSampler`, ensuring that a valid grasp is used for picking up the banana.
- **Placement Calculation**: The placement pose is calculated based on the current pose of the potted meat can and adjusted for the
gripper offset, ensuring that the banana is placed directly on top of it.
- **Generalization**: The plan is general enough to handle different initial positions and orientations of the objects, as it
dynamically calculates the placement pose based on the initial state.

This approach ensures that the banana will always be stacked on top of the potted meat can, accounting for the gripper offset along
the z direction.
```

```
Thank you for pointing that out. Let's correct the plan to account for the z-offset when placing the second cube.
```python
def gen_plan(initial:RavenBelief, x_offset:float, y_offset:float, z_offset:float):
    plan = []

    # Pick up the green block
    green_block = initial.objects["object_6"]
    plan.append(Action("pick", green_block.pose.point))

    # Place the green block into the green bowl
    green_bowl = initial.objects["object_5"]
    x, y, z = green_bowl.pose.point
    plan.append(Action("place", [x + x_offset, y + y_offset, z + z_offset]))

    # Pick up the yellow block
    yellow_block = initial.objects["object_4"]
    plan.append(Action("pick", yellow_block.pose.point))

    # Stack the yellow block onto the green block
    green_block_new_pose = [x + x_offset, y + y_offset, z + z_offset + BLOCK_SIZE]
    plan.append(Action("place", [green_block_new_pose[0], green_block_new_pose[1], green_block_new_pose[2] + BLOCK_SIZE]))

    return plan

def gen_domain(initial:RavenBelief):
    return {
        "x_offset": ContinuousSampler(-0.02, 0.02),
        "y_offset": ContinuousSampler(-0.02, 0.02),
        "z_offset": ContinuousSampler(0.01, 0.03),
    }
```

### Explanation

1. **Pick up the green block**: The plan starts by picking up the green block (`object_6`).
2. **Place the green block into the green bowl**: The green block is then placed into the green bowl (`object_5`). The offsets
(`x_offset`, `y_offset`, `z_offset`) allow for some variability in the exact placement within the bowl.
3. **Pick up the yellow block**: The plan continues by picking up the yellow block (`object_4`).
4. **Stack the yellow block onto the green block**: Finally, the yellow block is placed on top of the green block, which is now in
the bowl. The height is adjusted by adding the `BLOCK_SIZE` to the z-coordinate of the green block's new position, ensuring the
yellow block is stacked correctly.

### Why the Function Will Always Satisfy the Goal

- The `x_offset`, `y_offset`, and `z_offset` parameters allow for slight variations in the placement of the green block within the
bowl, ensuring it fits regardless of minor positional differences.
- The yellow block is always placed directly on top of the green block, ensuring a stack is formed.
- The domain for the offsets is chosen to be small enough to ensure the blocks are placed correctly but large enough to allow for
some variability.

### Constraints and Potential Failures

- If the offsets are too large, the green block might not be placed correctly within the bowl, or the yellow block might not stack
properly on the green block.
- The domain for the offsets is chosen to minimize these risks, but if a failure occurs, the offsets can be adjusted to smaller
ranges.

This plan should achieve the goal of placing the green block into the bowl and then stacking the yellow block onto the green block,
given the initial state and the defined domain.
```

**"Move banana to centre"**

```
To account for the slight offset along the y direction when grasping the object, we can adjust the placement position accordingly.
We will add an additional parameter to the `gen_plan` function to specify the y offset and use it to correct the placement
position.

Here's the updated implementation:

```python
def gen_plan(initial: RavenBelief, grasp: RavenGrasp, y_offset: float):
    plan = []
    object_name = "object_0"  # The banana is identified as "object_0"
    plan.append(Action("pick", [object_name, grasp]))
    # Adjust the y position by the offset when placing the object
    adjusted_y = TABLE_CENTER[1] + y_offset
    plan.append(Action("place", [object_name, grasp, RavenPose(x=TABLE_CENTER[0], y=adjusted_y, z=0.02)]))
    return plan

def gen_domain(initial: RavenBelief):
    return {
        "grasp": GraspSampler(),
        "y_offset": ContinuousSampler(-0.05, 0.05)  # Allowing for a small offset range
    }
```

### Explanation:
1. **gen_plan Function**:
   - The function now takes an additional parameter `y_offset` to account for the slight offset along the y direction.
   - When placing the object, the y position is adjusted by adding the `y_offset` to the table center's y coordinate.

2. **gen_domain Function**:
   - The function returns a dictionary with two entries:
     - `grasp`: Sampled using the `GraspSampler`.
     - `y_offset`: Sampled using a `ContinuousSampler` with a range of -0.05 to 0.05, allowing for a small adjustment in the y
direction.

### Why This Will Always Satisfy the Goal:
- The plan still explicitly picks the banana and places it at the center of the table, but now with an adjustment for the y
offset.
- The `y_offset` parameter allows for correcting the placement position to account for the slight offset during grasping.
- The table center coordinates are fixed and known, ensuring the banana is always moved to the correct location with the necessary
adjustment.

This approach ensures that the goal is achieved regardless of the specific grasp sampled and the slight offset during grasping, as
long as the offset is within the specified range.
```

Figure 13: **Examples of modified task-plan via AHA**

generation, and sub-task success detection. However, AHA is not yet capable of directly influencing low-level trained policies through failure language reasoning. A potential next step would involve training low-level policies with language-conditioned demonstrations covering a diverse range of corrective actions tied to various failure modes. This would enable low-level policies to interpret failure reasoning in a counterfactual manner and generate corrective actions directly.

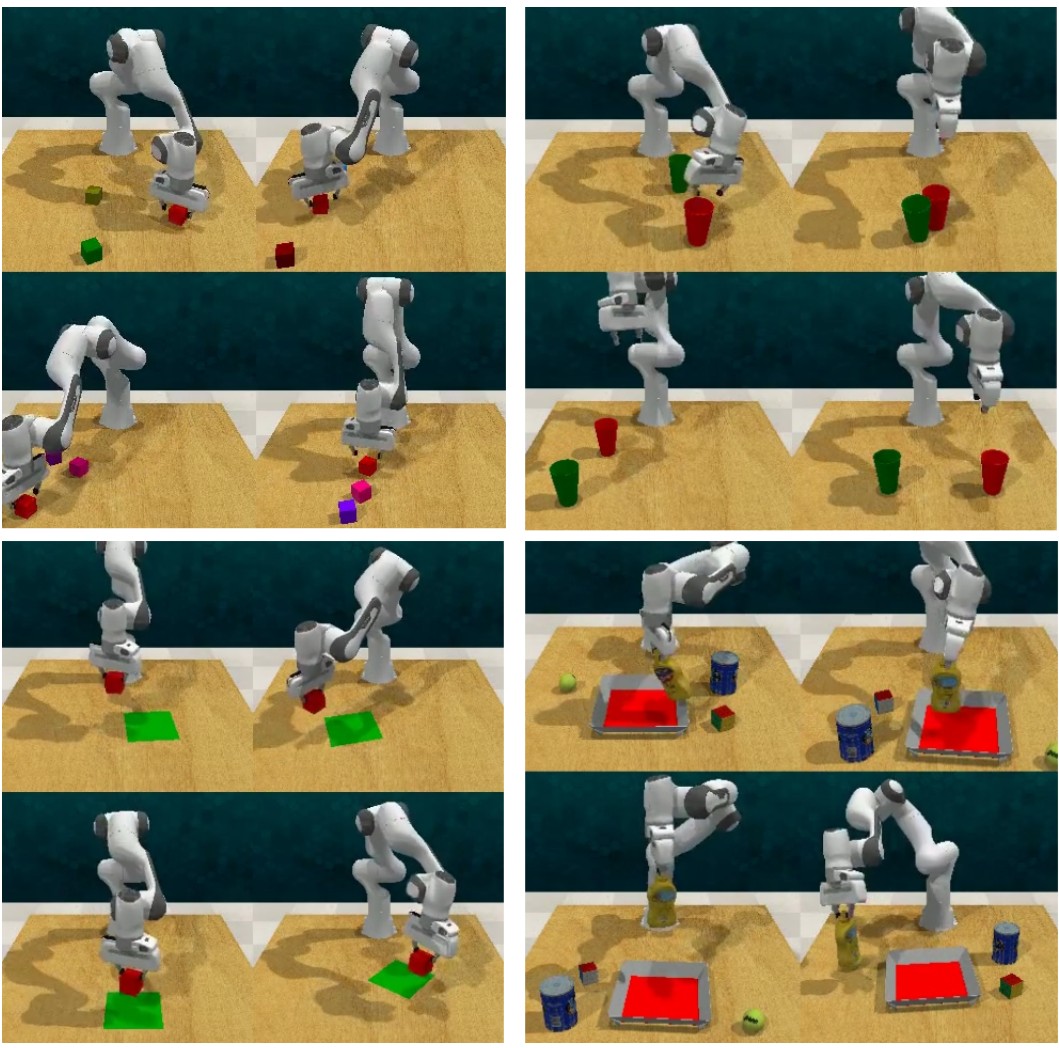

Figure 14: **Examples of zero-shot data generator trajectories with AHA as sub-tasks verifier.**
Row 1: `pickup_cube`, `pickup_cup`. Row 2: `put_block`, `sort_mustard`

