# OpenReview forum: "AHA: A Vision-Language-Model for Detecting and Reasoning Over Failures in Robotic Manipulation"
_ICLR.cc/2025/Conference — ICLR 2025 Poster_

### Official Review · Reviewer_qHQf · 2024-10-17

**Soundness:** 3
**Presentation:** 3
**Contribution:** 3
**Rating:** 8
**Confidence:** 5

**Summary:**

The paper presents a Vision-Language Model (VLM) for robotic failure reasoning, introducing a failure dataset and fine-tuning the VLM to achieve this goal. The approach demonstrates promising performance.

**Strengths:**

Failure trajectory generation can be extended to other scenarios (simulatores), encompassing a comprehensive set of failure categories. Failure reasoning plays a critical role in downstream policy learning, which is essential for advancing the entire community

**Weaknesses:**

Baseline Comparisons:
 - Clarity of Baseline Implementation: The implementation details of the baselines are unclear. What type of input data are the baselines using? Is it a single image, a sequence of images, images from multiple views, or video? This distinction significantly affects baseline performance. For instance, Gemini excels at reasoning for robotic manipulation when using video inputs, but its performance would likely degrade if multiple-view images were used instead.

 - Lack of Comparison with Robotic-Specific Methods: Why not compare your approach with other robotic-oriented reasoning methods mentioned in the 'Introduction' or 'Related Work' sections? The current experiments only compare against general Vision-Language Models (VLMs), which are not specifically designed for robotic tasks.

I would consider raising my score if these issues are addressed or clarified. Otherwise, it's difficult to properly assess how well the model performs.

**Questions:**

- View Angle Concerns: The camera view angles during training are fixed according to the RLBench settings. This raises concerns about the view angles during testing. Specifically: 1) How many views does the model require during testing? 2) Should the view angles during testing match the exact poses used in the RLBench training setting?

- Missing Reference to Figure 1: Figure 1 is not referenced or discussed anywhere in the paper, which needs to be addressed for clarity and completeness.

---

> ### Author Response · Authors · 2024-11-20
> **Rebuttal for Reviewer qHQf [1/2]**
>
> [Part 1/2]
>
> We thank the reviewer for their feedback. We appreciate your recognition of AHA as an approach that can be scaled up to other scenarios, and for acknowledging the crucial role that failure reasoning plays in downstream policy learning, which could be essential for advancing the field.  **All changes in the revised paper and supplementary materials have been highlighted in blue for easy reference.**
>
> Below, we address your comments, feedback, and clarification questions:
>
> **1. Clarity in implementation**
>
> >The implementation details of the baselines are unclear. What type of input data are the baselines using? Is it a single image, a sequence of images, images from multiple views, or video? This distinction significantly affects baseline performance. For instance, Gemini excels at reasoning for robotic manipulation when using video inputs, but its performance would likely degrade if multiple-view images were used instead.
>
> For all the proprietary and open-source models evaluated, we ensure consistency by using the same input images and prompt structure, as illustrated in Table 2 of the Supplementary Material. The layout of concatenated frame images varies across datasets due to differences in the number of views and other dataset-specific factors. For AHA, the input includes a sequence of images from three distinct views, with the sequence length varying depending on the task. Additionally, Maniskill-Fail uses three views, while RoboFail relies on a single view. While we recognize that different models may perform better with specific types of inputs, our primary objective is to enable a fair comparison by providing all models with the same contextual information.

---

> ### Author Response · Authors · 2024-11-20
> **Rebuttal for Reviewer qHQf [2/2]**
>
> [Part 2/2]
>
> **2. Additional baseline and ablation studies**
>
> > Lack of Comparison with Robotic-Specific Methods: Why not compare your approach with other robotic-oriented reasoning methods mentioned in the 'Introduction' or 'Related Work' sections? The current experiments only compare against general Vision-Language Models (VLMs), which are not specifically designed for robotic tasks.
>
> We appreciate the suggestion to compare our work with the studies mentioned in the related section. Many of these works utilize off-the-shelf Vision-Language Models (VLMs) or Large Language Models (LLMs), prompting them with task scene state information to perform failure detection and reasoning. This approach differs fundamentally from AHA, which is the first VLM specifically instruction-tuned for failure detection and reasoning. To address this suggestion, we implemented REFLECT, a failure reasoning system that also leverages off-the-shelf LLMs for reasoning. However, REFLECT incorporates state information obtained from a hierarchical scene graph constructed using multimodal sensory inputs. Despite these fundamental differences, we evaluated AHA and REFLECT on the RoboFail subset and found that AHA outperformed REFLECT in three out of four evaluation metrics, including binary success, LLM fuzzy matching, and cosine similarity.
>
>
> | **Approaches (Subset of RoboFail)** | **Binary Success** | **ROUGE-L** | **LLM Fuzzy Match** | **Cosine Similarity** |
> |----------------------------------|--------------------|--------------|----------------------|-----------------------|
> | **REFLECT**               | 0.677               | 0.352        | 0.637               | 0.305                 |
> | **AHA-13B**            | 0.733             | 0.178        | 0.754               | 0.354                 |
>
> >View Angle Concerns: The camera view angles during training are fixed according to the RLBench settings. This raises concerns about the view angles during testing. Specifically: 1) How many views does the model require during testing? 2) Should the view angles during testing match the exact poses used in the RLBench training setting?
>
> We thank the reviewer for raising this point. While AHA is trained on RLBench settings, our results in the main paper Table.2  demonstrate that the model can effectively reason in test scenarios where the viewing angle differs from those in RLBench and further supported with more ablation studies in supplementary material section 1.5. Additionally, we evaluated the AHA-trained model using different numbers of viewpoints during inference. The performance, measured by ROUGE, Fuzzy Matching, and Cosine Similarity scores, remained consistent whether 1, 2, or 3 views were used. This robust generalization likely stems from the reasoning capabilities inherent in the co-training data.
>
>
> | **Model: AHA-13B (Viewpoints)** | **Binary Success** | **ROUGE-L** | **LLM Fuzzy Match** | **Cosine Similarity** |
> |----------------------------------|--------------------|--------------|----------------------|-----------------------|
> | **One viewpoint**               | 1.000              | 0.673        | 0.587               | 0.712                 |
> | **Two viewpoints**              | 1.000              | 0.615        | 0.587               | 0.671                 |
> | **Three viewpoints**            | 1.000              | 0.600        | 0.633               | 0.681                 |
>
> >Missing Reference to Figure 1: Figure 1 is not referenced or discussed anywhere in the paper, which needs to be addressed for clarity and completeness.
>
> Thanks to the reviewer for pointing this out. We have added a reference to Figure 1 on line 103 of the paper.
>
> **We sincerely hope that by addressing the reviewer's concerns and providing the necessary clarifications, our paper may be considered for an improved rating. If there are any further questions, please do not hesitate to let us know. Thank you for your time and thoughtful review.**

---

> ### Author Response · Authors · 2024-11-22
> **Response to rebuttal response [Reviewer qHQf]**
>
> Dear Reviewer qHQf,
>
> Thank you for your thoughtful feedback and for recognizing the significance of failure detection and reasoning. As the rebuttal phase concludes, we kindly invite you to review our response. We have addressed your comments by:
>
> - Improving clarity in writing,
> - Adding experiments with REFLECT as a new robotics-specific baseline,
> - Conducting ablation studies on how the number of viewpoints impacts AHA's performance.
>
> We hope these updates address your concerns. If so, we would greatly appreciate it if you could consider raising our scores.
>
> Thank you again for your valuable feedback and support.

---

> > ### Comment · Reviewer_qHQf · 2024-11-24
> >
> > Thank you for your clear response, which has resolved my concern. Therefore, I will maintain my score as "Accept."

---

> > > ### Author Response · Authors · 2024-11-24
> > > **Thanks for the response, we welcome any further suggestions**
> > >
> > > Thanks for the prompt response to our rebuttal!
> > >
> > > The reviewer mentioned earlier in the initial review that *"if you address my concerns, I will be inclined to raise my score."* We welcome any additional feedback from the reviewer based on our added experiments, and encourage the reviewer to consider re-evaluating our work given the added content.

---

> > > > ### Author Response · Authors · 2024-11-26
> > > > **Any further questions [Reviewer qHQF]**
> > > >
> > > > We sincerely thank Reviewer qHQF for their response and are glad to have addressed all concerns clearly.  We also like to thank the reviewer for acknowledging failure reasoning plays a critical role in downstream policy learning, which is essential for advancing the entire community
> > > >
> > > > If there is further questions on the paper, please let us know. If not we kindly hope the reviewer will consider raising our rating as noted in the initial review. Thank you.

---

### Official Review · Reviewer_FdiA · 2024-11-01

**Soundness:** 2
**Presentation:** 3
**Contribution:** 3
**Rating:** 6
**Confidence:** 3

**Summary:**

This work proposes a VLM for detecting and reasoning about failures in robotic manipulation, which can be applied in various downstream tasks.

**Strengths:**

1. The motivation is meaningful, and I agree that addressing failures in robotics is important.
2. It can be utilized in downstream tasks.
3. The website has many great demos.

**Weaknesses:**

I think the types of failure modes and the reasoning responses are somewhat rough. And more details and experiments need to be conducted. For further details, please refer to the "Questions" section.

**Questions:**

1.What is the difference between rotation and no rotation?
2.What are the details of your baseline settings? I couldn't find them in your paper.
3.Are you using the REFLECT framework for your baseline? If not, I believe REFLECT is highly relevant to your work, so I would like to see a comparison of its performance with your model.
4.How do you plan to extend your failure types? As of now, I still find them somewhat rough.

My main concern is the sufficiency of the experiments and the work still seems somewhat preliminary. However, if you address my concerns, I will be inclined to raise my score.

---

> ### Author Response · Authors · 2024-11-20
> **Rebuttal for Reviewer FdiA**
>
> We thank the reviewer for their feedback. We appreciate the recognition of AHA as an interesting idea and see novelty in our approach of using AHA’s failure detection and reasoning to improve VLM-based robotics applications. Furthermore, we are grateful that the reviewer notes the importance of our AHA datasets for building robust large models that can better understand the physical world. **All changes in the revised paper and supplementary materials have been highlighted in blue for easy reference.**
>
> Below, we address your comments, feedback, and clarification questions:
>
> **1. Question regarding details in the paper**
>
> >What is the difference between rotation and no rotation? What are the details of your baseline settings? I couldn't find them in your paper.
>
> We acknowledge that our phrasing may have caused some confusion. The two failure modes differ as follows:
>
> **Incorrect Rotation Failure**
> This is an active failure mode where the robot reaches a keypoint with a rotation offset, which eventually leads to task failure.
>
> **Missing Rotation Failure**
> This is a more passive failure mode, where the robot is expected to adopt a new rotation at a keypoint but instead retains the rotation from the previous keypoints, resulting in failure.
>
> To address this, we have rephrased the wording in Section 3.1 for improved clarity. Additionally, as we evaluate our VLM’s capabilities alongside both open-source and proprietary VLMs, we have aligned the input prompts and system configurations for consistency. For more details on this alignment, please refer to Table 2 in the supplementary material.
>
> **2. Additional Baseline**
>
> >REFLECT seems highly relevant to the work, and would like to see it as a baseline.
>
> We thank the reviewer for suggesting the implementation of REFLECT as a baseline. In response, we implemented REFLECT on a subset of the RoboFail evaluation dataset and compared it with our method, AHA. Due to REFLECT’s reliance on multimodal sensory inputs such as audio—which are not available for other evaluation datasets—we limited the comparison to this subset. It is important to note the inherent differences between the two approaches. Our method involves instruction-tuning a Vision-Language Model (VLM) for failure detection and reasoning, while REFLECT uses a visual prompting technique that leverages off-the-shelf VLMs. REFLECT relies on detailed task and state information extracted from a hierarchical scene graph, which is constructed using diverse multimodal sensory inputs. Despite these differences, we evaluated AHA and REFLECT on the RoboFail subset and observed that AHA outperformed REFLECT in three out of four evaluation metrics, including binary success, LLM fuzzy matching, and cosine similarity.
>
> | **Approaches (Subset of RoboFail dataset)** | **Binary Success** | **ROUGE-L** | **LLM Fuzzy Match** | **Cosine Similarity** |
> |----------------------------------|--------------------|--------------|----------------------|-----------------------|
> | **REFLECT**               | 0.677               | 0.352        | 0.637               | 0.305                 |
> | **AHA-13B**            | 0.733             | 0.178        | 0.754               | 0.354                 |
>
> **3. Future plans for scaling AHA**
>
> >How can this work be scaled up and extended beyond its existing failure type.
>
> We acknowledge the current limitations of our approach to failure generation, which is conducted in a systematic manner focused on a predefined set of failure modes observed from existing robot datasets and policy rollouts. This method, while effective, does not encompass all potential failure modes and edge cases. As part of our future work, we plan to expand this approach through various methods, such as systematically distilling failures from policies using a real-to-sim setup or extracting latent representations of failures from real-world human videos. This enhancement seeks to improve the generalizability of our method beyond its current scope, enabling it to capture a broader range of failure scenarios.
>
> **We sincerely hope that by addressing the reviewer's concerns and providing the necessary clarifications, our paper may be considered for an improved rating. If there are any further questions, please do not hesitate to let us know. Thank you for your time and thoughtful review.**

---

> ### Author Response · Authors · 2024-11-22
> **Response to rebuttal response [Reviewer FdiA]**
>
> Dear Reviewer FdiA,
>
> Thank you sincerely for your time and thoughtful feedback on our paper. As the rebuttal phase nears its conclusion, we kindly invite you to review our submitted response. We have worked diligently to address your comments, including:
>
> - Clarifying terminology used in the paper,
> - Adding a new baseline with REFLECT and comparing results with AHA,
> - Expanding on future directions and potential extensions of AHA.
>
> We hope these revisions address your concerns thoroughly. If you find the updates satisfactory, we would be grateful if you could consider raising our scores.
>
> Thank you again for your valuable feedback and support.

---

> > ### Comment · Reviewer_FdiA · 2024-11-25
> >
> > Thank you for your response. But I still have the question that why you only test the subset of the RoboFail Dataset?

---

> > > ### Author Response · Authors · 2024-11-25
> > > **Response to the reviewer FdiA**
> > >
> > > We thank the reviewer for the response and the follow-up question. We tested only a subset of the RoboFail Dataset to minimize reliance on modalities like audio, which AHA is not designed to handle, while preserving the full implementation of REFLECT, as described in its paper. In future work, we plan to curate a REFLECT-style dataset to evaluate AHA's failure reasoning in real-world scenarios.
> > >
> > > An example of data point from within the RoboFail subset that could be used for reasoning about its failure without the need to  depend on audio input is:
> > >
> > > **putAppleBowl1**,
> > >
> > > ***gt_failure_reason***: "Apple is placed on top of the bowl instead of inside the bowl due to the bowl being upside down.",
> > >
> > > ***gt_failure_step***: ["01:00", "01:09"]
> > >
> > > **audio abnormally detected sequence**:  ["00:56", "00:58"]
> > >
> > > This data point has its audio sequence outside of the gt_failure_steps, which indicates to us that audio is not the main contributor to fail reasoning taken in by REFLECT, hence making it a good evaluation data point to be compared with AHA.
> > >
> > > while an example of a data point that was filter out of the evaluation subset for comparing REFLECT against AHA is which heavy depending its failure reasoning on audio is :
> > >
> > > **sauteeCarrot3**,
> > >
> > > ***gt_failure_reason***: "The robot accidentally dropped the knife when trying to slice the carrot.",
> > >
> > > **gt_failure_step**: "00:55"
> > >
> > > **audio abnormally detected sequence**:  00:54
> > >
> > > This provides the counter supporting point towards why we use a subset of RoboFail dataset.

---

> > > > ### Comment · Reviewer_FdiA · 2024-11-25
> > > >
> > > > Thank you for your response. I will raise my score to 6 for the interesting idea and motivation, but still I think a lot work you mentioned should be conducted in future like processing the audio information in text etc.

---

### Official Review · Reviewer_Enxm · 2024-11-03

**Soundness:** 2
**Presentation:** 3
**Contribution:** 4
**Rating:** 5
**Confidence:** 3

**Summary:**

This paper introduces AHA, a new VLM model to identify and describe failure modes in robot manipulation. The authors introduce a pipeline called FailGen to generate failure demonstrations and use the data to train AHA. In the experiments, AHA demonstrates improved performance in failure reasoning. The authors also show its effectiveness in downstream application including VLM-based reward design, planning, and subtask verification.

**Strengths:**

1. The idea of using failure detection VLM to improve VLM-based robotic application is interesting.
2. The effort in developing a failure dataset is valuable towards building robust large models that can understand the physical world.

**Weaknesses:**

1. [Soundness] It remains unclear to me whether incorporating failure descriptions is a reliable approach to improve VLM in reward design, planning, and subtask verification. For example, in the VLM-based reward design, although VLM can provide some penalty reward terms based on AHA input to correct behavior, there is no guarantee the new reward can make robot perform better than the previous one (e.g. the penalty can be too large and hinder skill learning). In this case, I would like to see if the proposed method can show some iterative reward tuning behavior based on failure feedback and the procedure can converge (i.e., tuning the strength of new terms or number of terms to correct behavior).

2. [Clarity] The paper can be much clearer and convincing if the authors can show more extensive comparisons between AHA and GPT4 to understand what leads to the improvements. For example, is the improvement due to GPT4 not being able to generate accurate failure description? Or, is it simply because the generated failure text is too lengthy for the downstream VLM model to understand? I encourage more explicit discussion on what a good failure description looks like (for downstream task VLMs). I also suggest comparing to human oracles with different writing styles.

3. [Result: Generalization] The paper is focusing on simple pick-and-place style tasks in clean backgrounds. How about more challenging and dexterous tasks? I would like to see the results on more challenging task such as guiding a robot hand to perform grasping and reorientations where the failure modes can be more fine-grained and complex. Besides, explicitly discussing generalization to diverse backgrounds / objects is also necessary. The results on these harder setups can help readers to understand the opportunities and challenges of AHA in application.

**Questions:**

See the weaknesses part. The idea is interesting and I would like to raise my score if the authors can show some rigorous results to clarify the points above, especially 1 and 2. I also encourage discussion on negative results to help readers see the limitation and potential of the current version.

---

> ### Author Response · Authors · 2024-11-20
> **Rebuttal for Reviewer Enxm**
>
> We thank the reviewer for their feedback. We appreciate the recognition of AHA as an interesting idea and see novelty in our approach of using AHA’s failure detection and reasoning to improve VLM-based robotics applications. Furthermore, we are grateful that the reviewer notes the importance of our AHA datasets for building robust large models that can better understand the physical world. **All changes in the revised paper and supplementary materials have been highlighted in blue for easy reference.**
>
> Below, we address your comments, feedback, and clarification questions:
>
> **1. Soundness in approach**
>
> >How can we guarantee that AHA improves some of these downstream applications, such as VLM-based reward design? Is it possible to observe iterative reward tuning behaviours as a result of failure feedback?
>
> We appreciate the reviewer’s question. While we do not claim that AHA will definitively improve downstream applications such as VLM-based reward design, we propose that failure reasoning modules like AHA could potentially benefit such applications. This is supported by the empirical experiments presented in the paper. It is important to note that our primary goal is not to provide a complete solution for reward design using failure feedback and reasoning but rather to demonstrate that this is a viable use case for AHA. To further illustrate this, we have included examples of AHA's application in iterative reward tuning for VLM-based reward design on the anonymous project page (https://aha-iclr.github.io/) for reference. Based on our observations, when AHA consistently succeeds in accurately reasoning about failures for the given tasks, it effectively guides the RL policy toward the optimal dense reward function.
>
> **2. Clarity in writing**
>
> >It would be more convincing to show more extensive comparisons of GPT4o against AHA. This would help in understanding if GPT4o performs poorer than AHA due to the context length of its output or other potential factors. One way to analyze that is to have a human oracle upper bound.
>
> **Generalization in approach**
>
> We apologize for the lack of clarity regarding the downstream tasks in our comparison between AHA and GPT-4o. Our experiments indicate that GPT-4o's poor performance was not due to the length of the failure descriptions but rather its inaccurate failure reasoning, which negatively impacted downstream task performance. To address this, we included example GPT-4o outputs for failure reasoning in the supplementary material Table 4. Additionally, based on the reviewer's feedback, we incorporated human-oracle evaluations with diverse writing styles. We recruited participants to curate and evaluate failure detection and reasoning outputs on the RoboFail dataset. These outputs were assessed using the four evaluation metrics, providing an upper performance bound for comparison with AHA. Although there is an obvious gap between AHA and human-oracle, AHA is still the SOTA in failure reasoning in comparison with other approaches.
>
>
> | **RoboFail Dataset** | **Binary Success** | **ROUGE-L** | **LLM Fuzzy Match** | **Cosine Similarity** |
> |----------------------------------|--------------------|--------------|----------------------|-----------------------|
> | **Human Oracle**               | 0.929              | 0.768        | 0.929               | 0.758                 |
> | **AHA-13B**              | 0.643             | 0.280        | 0.465               | 0.471                 |
>
> > How to make AHA more generalizable for other forms of robotics task reasoning?
>
> While we acknowledge that most downstream tasks in this paper focus on tabletop manipulation due to the nature of the fine-tuning data used for VLM training, we have tested AHA across various embodiments, viewpoints, tasks, and domains—even using randomly sampled manipulation videos from the internet for failure reasoning. Generating failure trajectories from simulation, one of our main contributions, and this is broadly applicable across domains. This data generation framework can be extended to domains such as locomotion, navigation, and dexterous manipulation, which we plan to explore in future work. The key challenge lies not in the methodology for failure generation and VLM fine-tuning presented in this work, but in adapting the failure dataset generation process to support broader domains. We are looking to expand this approach to systematically distill failures from policies using a real-to-sim setup, aiming to enhance the generalizability of our method beyond its current scope and will leave it for future work.
>
> **We sincerely hope that by addressing the reviewer's concerns and providing the necessary clarifications, our paper may be considered for an improved rating. If there are any further questions, please do not hesitate to let us know. Thank you for your time and thoughtful review.**

---

> ### Author Response · Authors · 2024-11-22
> **Response to rebuttal response [Reviewer Enxm]**
>
> Dear Reviewer Enxm,
>
> Thank you for your time, thoughtful feedback, and for recognizing the novelty and interest in our work. As the rebuttal phase concludes, we kindly invite you to review our response. We have worked diligently to address your comments, including:
>
> - Enhancing clarity in writing,
> - Providing further insights on how AHA can be extended,
> - Demonstrating AHA's usefulness in iterative reward guidance scenarios,
> - Incorporating a human oracle to evaluate VLM sensitivity to context length.
>
> We hope these revisions thoroughly address your concerns. If you find the updates satisfactory, we would greatly appreciate it if you could consider raising your scores for our paper.
>
> Thank you again for your valuable feedback and support.

---

> > ### Comment · Reviewer_Enxm · 2024-11-25
> > **Thank you for your response.**
> >
> > I would like to thank the authors for their response. I also appreciate the authors' effort in comparing their method to human oracle on text/success matching performance, although my intention was actually to see the performance of human oracle in application scenarios (e.g. improving reward design). Since my main concern is about the effectiveness of introducing failure reasoning to VLM in downstream tasks and this part is still a bit lacking, I will keep my score as 5.

---

> > > ### Author Response · Authors · 2024-11-25
> > > **Response to reviewer**
> > >
> > > Thank you for your thoughtful feedback. We apologize for any misunderstanding of your concerns. Our perspective is that if the VLM can reason about failures as effectively as a human (using a human oracle for evaluation on various reasoning datasets), it should naturally contribute to improving downstream task performance. To validate this further, we can run additional experiments replacing AHA with a human evaluator to assess the potential improvement in downstream tasks. We also appreciate the reviewers' input on specific concerns regarding the effectiveness of introducing failure reasoning in VLMs for downstream tasks. If the concern relates to Figure 3 in the main paper, we have included a plot showing the empirical contribution of AHA to improving performance in each downstream task. We welcome further clarification to better address these concerns and improve the paper.

---

> > > ### Author Response · Authors · 2024-11-26
> > > **Response to Reviewer Enxm**
> > >
> > > We appreciate the reviewer's thoughtful feedback and are encouraged to hear their positive view on the potential of using failure detection VLMs to enhance VLM-based robotic applications and build robust models for understanding the physical world. As the rebuttal period concludes, we kindly ask if our responses have addressed the reviewer's concerns or if there are any additional questions or suggestions that could further improve the paper.

---

> ### Author Response · Authors · 2024-11-26
> **Additional human oracle on text for downstream task performance**
>
> We thank the reviewer for their feedback. To address the concerns thoroughly, we conducted an additional human oracle study on the downstream task (VLM reward design). In this study, human participants watched video rollouts after each iteration of training, provided natural language reasoning on the failure feedback, and grounded this reasoning to improve dense reward function generation. The results are as follows:
>
> | **Task**       | **GPT4o** | **AHA-13B** | **Human Oracle** |
> |----------------|--------------|--------------|--------------|
> | **Stack_cube**     | 0.00        | 0.60         | 0.50         |
> | **Push_T**     | 0.30        | 0.40         | 0.30         |
> | **Open_drawer**     | 0.30        | 0.40         | 0.65         |
> | **Pickup_YCB**     | 0.25         | 0.45         | 0.50         |
> | **Place_sphere**     | 0.00         | 0.15         | 0.10         |
>
>
> Based on these results, we observe performance comparable to AHA when a human user provides natural language reasoning to improve VLM reward design. However, this performance can vary depending on the complexity of the RL tasks. Additionally, our evaluation was conducted with a fixed number of iterations rather than until convergence, which can differ for each task. However, based on the result, we do see that better failure reasoning helped with improving downstream tasks in general.

---

> ### Author Response · Authors · 2024-11-29
> **Follow-Up on Additional Experiment for VLM-Based Reward Design Tasks (Reviewer Enxm)**
>
> Dear Reviewer Enxm,
>
> Thank you for your thorough review and valuable feedback, which have greatly helped us improve the paper. We also sincerely appreciate that you recognize the interest, novelty, and importance of our work. Our work presents several novel contributions to the field:
>
> 1. The procedural failure generation pipeline for producing labeled failure demonstrations for instruction-tuning Vision-Language Models (VLMs).
>
> 2. The first instruction-tuning of a VLM to detect and reason about failures, demonstrating superior performance compared to open-source and proprietary VLMs across tasks, embodiments, and domains.
>
> 3. A range of use cases illustrating the versatility and potential applications of our approach.
>
> We kindly ask you to consider these contributions in your evaluation. Additionally, we have conducted an additional experiment using a human oracle for the VLM-based reward design tasks which you have requested, we hope you could have a reconsideration of our work.
>
> Please let us know if there are any further questions or concerns we can address.
>
> Thank you for your time and thoughtful consideration.

---

> > ### Comment · Reviewer_Enxm · 2024-11-30
> > **Thank you for your response.**
> >
> > Thanks for the update. The new results are very interesting and valuable. Even when the VLM is provided with (groundtruth) human-generated failure descriptions, it still struggles to design rewards that achieve even an 80% success rate. This indicates that relying on failure descriptions to enhance a VLM's reward generation capability is inherently limiting. Moreover, AHA appears to have already surpassed human performance in reward design, suggesting that **the proposed approach cannot be further improved by incorporating seemingly higher-quality labels**.
> >
> > Given these points, I encourage the authors to show clear evidence that failure reasoning could be valuable for robotics, especially considering that the tasks in experiments are already solvable through imitation learning and hand-engineered RL rewards. While the dataset itself is a valuable contribution, its utility for various downstream applications remains uncertain. Without clear evidence of its potential impact in these areas, I will maintain my current score.

---

> ### Author Response · Authors · 2024-11-30
> **Response to Reviewer Enxm**
>
> We would like to thank the reviewer for the follow-up. We wanted to clarify a few points:
>
> > *Even when the VLM is provided with (ground-truth) human-generated failure descriptions, it still struggles to design rewards that achieve even an 80% success rate.*
>
> We would like to clarify that our VLM (AHA) does not directly design rewards for methods like EUREKA. Instead, AHA detects failures, reasons about them, and passes the insights to a proprietary LLM (GPT-4o) for reward design, as mentioned in L481-482 of the paper. Therefore, the results demonstrate that AHA enhances the success rate of downstream tasks leveraging EUREKA by providing more accurate failure reasoning, compared to the default EUREKA reflection mechanism (replaced with the proprietary VLM) as shown in Figure 3 (right) in the paper.
>
> > *This indicates that relying on failure descriptions to enhance a VLM's reward generation capability is inherently limiting.*
>
> We acknowledge that achieving a 100% success rate remains a challenge. However, we emphasize the significant 22.31% improvement over the baseline, demonstrating a substantial performance gain.
>
> > *Moreover, AHA appears to have already surpassed human performance in reward design, suggesting that the proposed approach cannot be further improved by incorporating seemingly higher-quality labels.*
>
> We recognize that reward design incorporating AHA's failure reasoning occasionally surpasses reward design using human feedback. However, we argue that humans should not be considered the upper bound for reward design. Effective reward shaping often involves more nuanced insights, as evidenced by the EUREKA paper. For example, in Figure 8 of their work, the authors show that purely human-based reward shaping performs poorly compared to LLM-aided approaches. This aligns with our findings, even across very different tasks in Eureka paper, whereby leveraging a VLM like AHA could potentially achieve comparable or occasionally superior performance to manual human feedback.
>
> > *especially considering that the tasks in experiments are already solvable through imitation learning and hand-engineered RL rewards.*
>
> We respectfully disagree with the reviewer on this, both imitation learning and hand-engineering for RL rewards require lots of efforts from human experts in robotics to collect demonstrations or design dense rewards. AHA, combined with Eureka, remove the requirements for human expertise/human intelligence to solve robotic tasks hence making both work a contribution to the field.
>
> > *its utility for various downstream applications remains uncertain*
>
> We appreciate the reviewer acknowledging our contribution to failure reasoning as a dataset. However, we respectfully disagree with the claim that our contribution lacks utility for downstream tasks solely because these tasks are already addressed by existing methods. Firstly, our work extends beyond helping with improving VLM-based reward design, we also shown that AHA can help with improving LLM-based task planning for TAMP systems or enhanced task verification for zero-shot manipulation systems. With the increasing adoption of off-the-shelf LLMs and VLMs in robotics, AHA's failure reasoning and feedback have the potential to support a wide range of future tasks and downstream applications.
>
> **Given these clarifications, we kindly hope you will take them into consideration when re-assessing our work.**

---

> > ### Author Response · Authors · 2024-12-03
> > **Response to Reviewer Enxm**
> >
> > We sincerely appreciate the reviewer’s efforts in providing continuous feedback and thoughtful clarifications on our work. We are encouraged by their positive perspective on the potential of failure detection VLMs to enhance VLM-based robotic applications and to build robust models for understanding the physical world. As the rebuttal period concludes, we kindly ask if our responses and clarifications have adequately addressed the reviewer’s concerns or if there are additional questions or suggestions that could further improve the paper. If we have successfully demonstrated the potential of AHA as a valuable tool for advancing future VLM/LLM-based robotics systems through its failure reasoning capabilities, we humbly hope the reviewer might consider raising our scores.

---

### Official Review · Reviewer_zG7S · 2024-11-03

**Soundness:** 3
**Presentation:** 3
**Contribution:** 3
**Rating:** 6
**Confidence:** 3

**Summary:**

This work targets the task of open-world failure detection and reasoning under the scenario of robotic manipulation, which is critical and valuable for diverse downstream tasks. Specifically, the authors propose a procedural pipeline for generating failure demonstration data for robotic manipulation in simulation, upon which they fine-tune a vision-language model (VLM) for failure detection and reasoning. Moreover, they demonstrate improved performance brought by the fine-tuned VLM in three downstream tasks, namely reward function design, task-plan refinement, and sub-task verification. Comprehensive comparison with the other six VLMs also shows the superior performance of the proposed technique on this task.

**Strengths:**

- The paper tackles an important problem that is highly beneficial and safety-critical when learning-based robotic algorithms are deployed in the real world;

- The developed procedural pipeline of simulation data generation is useful for researchers in the community who are willing to contribute to this direction;

- It is inspiring to see that not only improved performance is shown on the standalone task, but to see the strong benefits brought by such improvement on different related and popular downstream tasks.

- Though failure detection is an important prerequisite for the reasoning part, I am still glad to see the efforts in investigating the failure reasoning and especially its usage in the downstream tasks.

- The presentation is generally well-structured and easy to follow.

- Though benchmarking experiments are conducted in the simulation, their results are comprehensive and informative.

**Weaknesses:**

Presentation:
- It would be clearer if subsection 4.1. can be shortened a bit and described with a more concise mathematical formulation.

- To reinforce the contribution, I would suggest adding a subsection in the method section to explain how improved failure detection and reasoning performance can help these downstream tasks. In the current draft, putting this part in the experiment section is less apparent and inspiring for potential readers in these fields.

- Closely related to the point above, it seems that the usage of the proposed idea in downstream task 2 (refine task plans in TAMP) and task 3 (improve task verification) is essentially similar, i.e., detect and reason for the failure of the sub-tasks. It might be more lucid for the readers to group these two tasks together along with the reward design task and make a taxonomy for different high-level manners of how failure detection and reasoning can help.

- in sub-section 4.1., "Unlike previous works that primarily focus on detecting task success as binary classification problem, we approach failure reasoning by first predicting a binary success condition ("Yes" or "No") of the given sub-task based on a language specification and an input image prompt." seems less straightforward to understand. Logically, I suspect the author wants to express that they have an additional reasoning part after the detection step.


- Fig.1 and Fig.2 are hard to read. It would be better to reorganize the sub-figures and increase the sizes of the images and texts.

Methodology:

- The first point I would like to raise is the lack of discussion of hallucination in LLM/VLMs. In the introduction, the authors motivate the failure recognition with the shortcoming of hallucinations in LLMs/VLMs. However, the proposed fine-tuned VLM will still suffer from this issue. It would be important to discuss how to mitigate this issue in detecting failures. In this regard, there is a missing citation [1] that is closely related to this work, which made an initial attempt in this direction based on the uncertainty derived from the VLM. I would suggest the authors add a discussion on this aspect. It would also be meaningful to see the influence of fine-tuning with the proposed data set on the performance of uncertainty estimation. Will there be a trade-off between the performance of the prediction accuracy and uncertainty estimation?

- The second point is related to the design of the technique. In [1], it has been observed that using the "chain-of-thoughts" (CoT) analysis in the prompt template can improve the performance of failure detection. Therefore, related to the reasoning part of this work, will the reasoning part be able to help improve the performance failure detection? Will the performance be influenced by the different prompting templates, e.g., with and w/o CoT?

Experiments:

- Due to missing real-world experiments, therefore, it is hard to evaluate the performance loss during sim-to-real transfer and the actual practicality of the proposed technique.


[1] "Zheng, Zhi, et al. "Evaluating Uncertainty-based Failure Detection for Closed-Loop LLM Planners." arXiv preprint arXiv:2406.00430 (2024)."

**Questions:**

Side questions (major ones in the Weaknesses section above)

- In subsection 3.1., why separate the rotation failure into incorrect and missing ones? Should these two belong to the same category?

- An ablation study on how many camera viewpoints would be insightful for the real-robot applicability of the proposed method.

---

> ### Author Response · Authors · 2024-11-20
> **Rebuttal for Reviewer zG7S [1/2]**
>
> [Part 1/2]
>
> We thank the reviewer for their thoughtful and detailed feedback, which reflects their significant effort. We appreciate the reviewer's recognition of the importance of our work for safety-critical applications when deploying learning-based robotics algorithms in the real world, and acknowledge the importance of failure detection and reasoning. We are pleased that the reviewer found our paper well-structured, easy to follow, and that our results are comprehensive and informative. We have made substantial efforts to revise the paper's presentation and methodology, and have included additional experiments as suggested by the reviewers, all aimed at improving the paper's overall quality. **All changes in the revised paper and supplementary materials have been highlighted in blue for easy reference.**
>
> Below, we address your comments, feedback, and clarification questions:
>
> **1. Clarity in writing**
>
> >It would be clearer if subsection 4.1. can be shortened a bit and described with a more concise mathematical formulation. Furthermore, adding a subsection in method to explain how improved failure detection and reasoning performance can help these downstream tasks.
>
> We thank the reviewer for the advice, and have revised Section 4.1 to include a more concise mathematical formulation. Furthermore, we have added Section 4.4 as a new subsection to provide more explanation on how AHA can help with these downstream tasks.
>
> >Fig.1 and Fig.2 are hard to read. It would be better to reorganise the sub-figures and increase the sizes of the images and texts.
>
> We thank the reviewer for the feedback on the figures and have made the necessary adjustments to Figures 1 and 2 to enhance their readability.
>
> **2. Questions on Methodology**
>
> >How does the proposed fine-tuned VLM address hallucination issues in failure detection, and what is the impact of fine-tuning with the dataset on uncertainty estimation and the trade-off with prediction accuracy? Additionally, there is missing citation.
>
> We thank the reviewer for the suggested citation. We have added the missing references and discussion in Lines 143–145, acknowledging that our method does not fully address hallucination in LLMs/VLMs. However, fine-tuning on domain-relevant data has shown improved performance in robotics scenarios. To highlight the importance of fine-tuning with our proposed dataset, we examined the correlation between AHA's prediction accuracy and its uncertainty estimations. Specifically, we compared sentence token prediction probabilities from AHA-13B with those from LLaVA v1.5-13B. AHA-13B demonstrated significantly higher average prediction probabilities, aligning with its superior accuracy. These findings, which indicate a positive impact of fine-tuning with the AHA failure dataset, are included in the supplementary material section 1.5.
>
> | Dataset              | AHA-13B (Output Probabilities / Cosine Similarity) | LLaVA-13B-v1.5 (Output Probabilities / Cosine Similarity) |
> |----------------------|----------------------------------------------------|----------------------------------------------------------|
> | **AHA (Test)**  | 0.670 / 0.583                                      | 0.0666 / 0.208                                           |
> | **Maniskill Fail**   | 0.457 / 0.681                                      | 0.024 / 0.208                                            |
> | **RoboFail**         | 0.292 / 0.471                                      | 0.000 / 0.203                                            |
>
>  >How does CoT prompting or other prompting templates help with the baselines so as to compare with AHA.
>
> We appreciate the reviewer’s suggestion that Chain-of-Thought (CoT) prompting or other advanced prompting methods could potentially improve baseline performance, such as GPT-4o, beyond the in-context learning results already presented in the original paper. To investigate this, we implemented Multimodal Chain-of-Thought Reasoning (MMCoT) [1] and evaluated it with GPT-4o on the RoboFail dataset. Despite these enhancements, our approach continues to outperform MMCoT on GPT-4o, demonstrating its effectiveness in failure detection and reasoning.
>
> [1] Zhang, Zhuosheng, Aston Zhang, Mu Li, Hai Zhao, George Karypis, and Alex Smola. "Multimodal chain-of-thought reasoning in language models." arXiv preprint arXiv:2302.00923 (2023).
>
> | **RoboFail Dataset** | **Binary Success** | **ROUGE-L** | **LLM Fuzzy Match** | **Cosine Similarity** |
> |----------------------------------|--------------------|--------------|----------------------|-----------------------|
> | **GPT4o+MMCoT**               |    0.571          |   0.097     |    0.444           | 0.307                |
> | **AHA-13B**              | 0.643             | 0.280        | 0.465               | 0.471                 |

---

> ### Author Response · Authors · 2024-11-20
> **Rebuttal for Reviewer zG7S [2/2]**
>
> [Part 2/2]
>
>
> **3. Additional Ablation Study**
>
> >It would be insightful to understand how varying the number of camera viewpoints affects downstream tasks.
>
> We thank the reviewer for suggesting an ablation study on the impact of camera viewpoints for failure reasoning in downstream applications. To empirically evaluate how viewpoints affect AHA's performance, we conducted zero-shot evaluations on the ManiSkill-Fail dataset. Specifically, we compared three input configurations: single-viewpoint images, concatenated two-viewpoint images, and the original dataset configuration with three concatenated viewpoints. Our results indicate a slight performance advantage when using single-viewpoint images. We attribute this to the resolution limitations of the LLaVA-1.5 visual encoder (256x256), where single-viewpoint inputs provide clearer visual information for failure reasoning. The ablation results are attached for reference:
>
> | **Model: AHA-13B (Viewpoints)** | **Binary Success** | **ROUGE-L** | **LLM Fuzzy Match** | **Cosine Similarity** |
> |----------------------------------|--------------------|--------------|----------------------|-----------------------|
> | **One viewpoint**               | 1.000              | 0.673        | 0.587               | 0.712                 |
> | **Two viewpoints**              | 1.000              | 0.615        | 0.587               | 0.671                 |
> | **Three viewpoints**            | 1.000              | 0.600        | 0.633               | 0.681                 |
>
> >Due to missing real-world experiments, therefore, it is hard to evaluate the performance loss during sim-to-real transfer and the actual practicality of the proposed technique.
>
> We acknowledge the limitation of not including real-world experiments in our work. This decision stems from the significant differences in setup across the three downstream tasks (Eureka, Trust the Process, and Manipulate-Anything). However, we provided quantitative results in simulation, conducted in the same manner as in the original papers for these tasks. Each of those papers includes its own demonstration of real-world experiments and the feasibility of sim-to-real transfer, supporting the applicability of our approach in real-world settings. The primary focus of AHA is to demonstrate how a Vision-Language Model (VLM) can be fine-tuned for failure detection and reasoning, showcasing its effectiveness and its potential for a wide range of downstream robotics applications. These applications are not limited to the three examples presented in the paper but extend to broader use cases in the field. Lastly, sim2real for these downstream tasks are not part of our contribution of the paper.
>
> **We sincerely hope that by addressing the reviewer's concerns and providing the necessary clarifications, our paper may be considered for an improved rating. If there are any further questions, please do not hesitate to let us know. Thank you for your time and thoughtful review.**

---

> ### Author Response · Authors · 2024-11-22
> **Response to rebuttal response [Reviewer zG7S]**
>
> Dear Reviewer zG7S,
>
> Firstly, we sincerely appreciate the time and effort you have dedicated to reviewing our paper. As the rebuttal discussion session draws to a close, we kindly request you to review our submitted rebuttal. We have made significant efforts to address your comments, including:
>
> - Reorganizing Section 4.1 and adding a subsection in the Methods section to clarify how improved failure detection and reasoning contribute to downstream tasks.
> - Incorporating the missing citation you highlighted and conducting additional experiments to analyze AHA's prediction likelihood versus accuracy, addressing the issue of uncertainty estimation in VLM predictions.
> - Performing further experiments using MM-CoT to evaluate the impact of prompting and reasoning on performance, particularly in comparison to closed-source models like AHA.
>
> We hope these revisions comprehensively address your concerns. If so, we would greatly appreciate it if you could consider raising our scores.
>
> Thank you once again for your valuable feedback and consideration.

---

> > ### Author Response · Authors · 2024-11-25
> > **Further questions from Reviewer zG7S**
> >
> > Dear Reviewer zG7S,
> >
> > Firstly, we sincerely appreciate the time and effort you have dedicated to reviewing our paper. As the rebuttal discussion session draws to a close, we kindly request you to review our submitted rebuttal. We have made significant efforts to address your comments, and we welcome any further questions that you might have.
> >
> > Thank you once again for your valuable feedback and consideration.

---

> > ### Comment · Reviewer_zG7S · 2024-11-26
> > **Feedback from Reviewer zG7S**
> >
> > Thank you for modifying the manuscript and conducting additional experiments using MM-CoT, uncertainty estimation, and multiple camera viewpoints.
> > However, regarding the experiments on uncertainty estimation, I am confused about understanding it. Does the consine similarity represent the success rate or accuracy? How about other metrics? The other model (LLaVA-13B-v1.5) also shows low likelihoods with low similarity, which means that it is also proportional to how likely the prediction can be wrong, right? In this regard, I don't see the "positive impact of fine-tuning with the AHA failure dataset" mentioned by the author.

---

> ### Author Response · Authors · 2024-11-26
> **Response to reviewer zG7S**
>
> We appreciate the reviewer’s feedback and apologize for any confusion. For uncertainty estimation experiments, we used two metrics:
>
> 1. **Cosine Similarity**: This measures the similarity between the ground-truth sentence and the model’s prediction by calculating the cosine of the angle between their vector representations. These vectors are typically derived using embedding methods such as TF-IDF, Word2Vec, GloVe, or modern techniques like BERT. Which the closer to 1, the cosine similarity the more accuracy is the model prediction. (Which was also used in the main paper as one of the key metric of evaluation, in Table 2)
>
> 2. **Sentence Token Likelihood**: This metric evaluates the probability of a sequence of tokens forming a sentence, as assigned by a language model. It reflects the model's confidence in its predictions. We compute this by first finding the model's individual token probability, and the overall likelihood of the sentence is calculated by multiplying the probabilities of all the tokens in the sequence.  The closer the score to 1, means the more confident the model is confidence about its prediction of the sentence.
>
> We use Cosine similarity as a way of measuring semantic meaning accuracy of the model in the paper, and in this additional experiment we introduce sentence token likelihood to measure confidence of the model's prediction. Hence, our results show that AHA outperforms LLaVA-13B-v1.5 in generating language outputs more similar to the ground truth (higher cosine similarity) while also exhibiting greater confidence (higher sentence likelihood). This demonstrates the positive impact of fine-tuning with the AHA dataset. We hope this clarifies our approach. Thank you.

---

> ### Comment · Reviewer_zG7S · 2024-11-26
> **Correlation between Likelihood and Similarity**
>
> Thank you for the explanation. If I understand correctly, you use the likelihood as the uncertainty/confidence measure. The desired relationship between the likelihood and the accuracy (similarity) should be linearly proportional, assessing how well
> the predicted uncertainty coincides with the prediction error, i.e., the higher the likelihood, the higher the accuracy, and vice versa. One example can be the Area Under Sparsification Error (AUSE) [1]. The current results don't express such a correlation as the underperformed model might have a higher correlation.
>
> [1] Lind, Simon Kristoffersson, et al. "Uncertainty quantification metrics for deep regression." Pattern Recognition Letters 186 (2024): 91-97.

---

> ### Author Response · Authors · 2024-11-26
> **Response to Reviewer zG7S**
>
> We thank the reviewer for their thoughtful comments and for engaging with our explanation. To clarify, we do not claim that the likelihood is perfectly correlated with accuracy (cosine similarity) or always correlated. Instead, we argue that there is a general trend where higher likelihood is associated with higher accuracy. This trend is observed in AHA, where predictions with higher likelihood tend to exhibit higher accuracy. Conversely, the baseline model demonstrates the opposite behavior, with lower likelihood correlating with lower accuracy.
>
> We acknowledge the importance of quantitatively evaluating this relationship and agree that metrics like Spearman correlation could provide a rigorous way to assess the alignment between likelihood and accuracy. We will consider incorporating such analyses to strengthen our evaluation and further illustrate the differences between AHA and the baseline model.

---

> > ### Author Response · Authors · 2024-11-30
> > **Response to Reviewer zG7S**
> >
> > Dear Reviewer zG7S,
> >
> > We sincerely thank you for your thoughtful suggestions and detailed feedback. Your insights have significantly helped us improve the quality of our work. We deeply appreciate your recognition of the importance of failure reasoning and your alignment with our ideas on its impact on downstream tasks. We are also delighted to hear that you enjoyed reading our paper.
> >
> > We apologize for any confusion in our initial interpretation of your comments. We have clarified our positions and claims in response and have addressed the points you raised. Additionally, your suggestions have inspired us to consider new directions for future work.
> >
> > Overall, we hope that our responses have addressed all your concerns and resolved any doubts about the paper. Your feedback has been invaluable in helping us refine and enhance our work, and we are genuinely grateful for your contributions to this process.

---

### Author Response · Authors · 2024-11-20
**General Rebuttal to All Reviewers**

We are grateful to the reviewers for their thorough efforts in reviewing the paper and providing us with thoughtful feedback, which has been invaluable in improving this work.

We appreciate the positive feedback highlighting that the paper is "well-motivated" and addresses failure reasoning, an "important" problem for the robotics community [zG7s, FdiA, qHQF]. We are grateful that reviewers found our problem formulation, writing, and demos to be clear, well-structured, and easy to follow [zG7S, FdiA]. The results were described as impressive, comprehensive, and informative [zG7S], with reviewers recognizing the novelty of our proposed approach [Enxm]. Additionally, many reviewers acknowledged the effectiveness of AHA in improving downstream robotics applications [Enxm, FdiA, zG7s].

Despite the positive feedback, reviewers have highlighted several areas for improvement, which we have summarized below:

1. **Clarity in Writing for Downstream Tasks** [zG7S, Enxm, FdiA,qHQF]
   - Reviewers emphasized the need for clearer articulation of how our approach supports downstream robotics tasks.
2. **Correlation Between AHA’s Performance and Prediction’s Uncertainty Estimation** [zG7S]
   - There is interest in further exploring the relationship between AHA’s model performance and its prediction uncertainty estimation.
3. **Comparison With Conventional Approaches** [FdiA, qHQF]
   - A deeper comparison between AHA and traditional failure reasoning methods, such as those proposed in REFLECT, was requested.
4. **Impact of Instruction-tuning Across Viewpoints** [qHQF, zG7S]
   - Reviewers noted the need for additional analysis on how instruction-finetuning with our proposed datasets affects performance when evaluated with varying numbers of viewpoints.
5. **Future Work on Generalization Across Domains** [Enxm, FdiA]
   - Suggestions were made to expand the discussion of future work, particularly how AHA could generalize to broader robotics domains such as locomotion, bimanual manipulation, and navigation.

In response to the reviewers' feedback, we have uploaded a revised version of the paper (revisions highlighted in blue text) with the following changes and conducted several additional experiments and ablation studies.

1. **[Improved Clarity on AHA’s Contributions](https://openreview.net/forum?id=JVkdSi7Ekg&noteId=NRk7OKWnDY)**
   - Section 4 method section has been revised, including a new subsection to better articulate AHA’s contributions to downstream tasks.

2. **[Prediction Uncertainty Analysis](https://openreview.net/forum?id=JVkdSi7Ekg&noteId=mAj8LuN6kr)**
   - Additional experiments have been conducted to compare AHA's average sentence output likelihood probabilities with those of LLaVA-v1.5-13B, providing new insights into AHA’s prediction uncertainty.

3. **[Comparison with REFLECT](https://openreview.net/forum?id=JVkdSi7Ekg&noteId=Zjm2CDYIbP)**
   - REFLECT has been implemented on a subset of the RoboFail evaluation dataset to enable a direct empirical comparison with AHA.

4. **[Ablation Studies on Viewpoints](https://openreview.net/forum?id=JVkdSi7Ekg&noteId=BtK7x2GJvA)**
   - We conducted further ablation studies to analyze the impact of varying numbers of viewpoints on AHA’s performance during evaluation.

5. **[Expanded Discussion on Limitations and Future Directions](https://openreview.net/forum?id=JVkdSi7Ekg&noteId=Zjm2CDYIbP)**
   - Section 6 conclusion section and supplementary material have been revised to include a more detailed discussion of AHA’s limitations and potential future work, including ways to extend AHA to other domains.


**We sincerely hope that by addressing all of the reviewers' concerns and providing the necessary clarifications, our paper may be considered for an improved rating. If there are any further questions, please do not hesitate to let us know. Thank you for your time and thoughtful review.**

---

### Meta-Review · Area_Chair_J4TY · 2024-12-18

**Metareview:**

The paper introduces AHA, a vision-language model for detecting and reasoning about failures in robotic manipulation tasks. The method demonstrates promising performance in analyzing manipulation failures and improving downstream robotic applications, receiving generally positive reviews across all reviewers (scores 5-8). Reviewers consistently praised the paper's meaningful motivation of addressing failure reasoning in robotics, its novel contribution of a failure dataset generation framework, and the demonstrated utility in multiple downstream applications including reward design, planning, and verification tasks.

Initial concerns centered on three main areas: the lack of rigorous evaluation and comparison with robotics-specific baselines like REFLECT, unclear implementation details about input modalities and baseline configurations, and questions about generalization beyond simple pick-and-place scenarios. The authors provided detailed responses, implementing new comparisons with REFLECT that showed AHA's superior performance, clarifying their standardized input protocol across baselines, and conducting additional ablation studies on viewpoint variations. They also demonstrated comparable performance to human oracle evaluations in some cases.

While two reviewers were satisfied with these responses and maintained positive scores, one reviewer remained concerned about the fundamental limitations of using failure descriptions for improving downstream tasks, noting that even with human-generated descriptions the success rates remained below 80%. However, the authors clarified that AHA serves to enhance existing frameworks rather than replace them entirely, showing consistent improvements over baselines across different applications.

**Additional Comments On Reviewer Discussion:**

None -- see metareview

---

### Decision · Program_Chairs · 2025-01-22

Accept (Poster)